# Cryo-EM structure of respiratory complex I at work

**Kristian Parey[1], Ulrich Brandt[2,3], Hao Xie[4], Deryck J Mills[1], Karin Siegmund[5,6], Janet Vonck[1], Werner Kühlbrandt[1,3], Volker Zickermann[3,5,6]***

[1]Department of Structural Biology, Max Planck Institute of Biophysics, Frankfurt, Germany; [2]Radboud Institute for Molecular Life Sciences, Department of Pediatrics, Radboud University Medical Centre, Nijmegen, The Netherlands; [3]Cluster of Excellence Macromolecular Complexes, Goethe University Frankfurt, Frankfurt, Germany; [4]Department of Molecular Membrane Biology, Max Planck Institute of Biophysics, Frankfurt, Germany; [5]Medical School, Institute of Biochemistry II, Goethe University Frankfurt, Frankfurt, Germany; [6]Centre for Biomolecular Magnetic Resonance, Institute for Biophysical Chemistry, Goethe University Frankfurt, Frankfurt, Germany

**Abstract** Mitochondrial complex I has a key role in cellular energy metabolism, generating a major portion of the proton motive force that drives aerobic ATP synthesis. The hydrophilic arm of the L-shaped ~1 MDa membrane protein complex transfers electrons from NADH to ubiquinone, providing the energy to drive proton pumping at distant sites in the membrane arm. The critical steps of energy conversion are associated with the redox chemistry of ubiquinone. We report the cryo-EM structure of complete mitochondrial complex I from the aerobic yeast *Yarrowia lipolytica* both in the deactive form and after capturing the enzyme during steady-state activity. The site of ubiquinone binding observed during turnover supports a two-state stabilization change mechanism for complex I.

DOI: https://doi.org/10.7554/eLife.39213.001

***For correspondence:**
Zickermann@med.uni-frankfurt.de

## Introduction

Respiratory complex I is a ~1 MDa membrane protein complex with key functions in aerobic energy metabolism (*Hirst, 2013*; *Wirth et al., 2016*). Fourteen central subunits are conserved from bacteria to mammals. Mitochondrial complex I contains in addition around 30 accessory subunits. The energy released in the electron transfer reaction from NADH to ubiquinone is utilized to pump protons across the inner mitochondrial membrane or the bacterial cell membrane. The electrochemical gradient established by redox-coupled proton translocation drives ATP synthesis. Dysfunction of complex I is implicated in a number of neuromuscular and degenerative diseases (*Rodenburg, 2016*). In myocardial infarction, complex I releases detrimental reactive oxygen species (ROS) that contribute to reperfusion injury (*Chouchani et al., 2013*). The reversible conversion of the active A-form of complex I into the deactive D-form (A/D transition) (*Kotlyar and Vinogradov, 1990*) is thought to minimize ROS formation.

X-ray structures of bacterial complex I (*Baradaran et al., 2013*) and of the mitochondrial complex I from the aerobic yeast *Yarrowia lipolytica* (*Zickermann et al., 2015*) were solved at 3.3 and 3.6 to 3.9 Å resolution, respectively. Recent technical progress has permitted the structure of mammalian complex I by itself or as part of a supercomplex to be determined at 3.3 to 4.2 Å resolution by electron cryo-microscopy (cryo-EM) (*Zhu et al., 2016*; *Fiedorczuk et al., 2016*; *Gu et al., 2016*; *Wu et al., 2016*; *Letts et al., 2016*; *Guo et al., 2017*; *Blaza et al., 2018*; *Agip et al., 2018*). Even though the structure of complex I is now well-characterized, essential mechanistic details of

the catalytic cycle remain elusive, and conflicting models for the A/D transition have recently been proposed (*Zickermann et al., 2015*; *Blaza et al., 2018*). There is general agreement that the reduction of ubiquinone at a position above the membrane bilayer triggers and drives proton translocation, but it is not clear how this works. We hypothesized that the power stroke generated during ubiquinone reduction results from a concerted rearrangement of its binding site and the surrounding protein region, driven by the stabilization of anionic ubiquinone intermediates (*Zickermann et al., 2015*). This mechanism implies a cycling between two alternating ubiquinone binding sites during turnover, one of which (the E-state) selectively allows reduction and the other (the P-state), protonation of the ubiquinone headgroup. The associated conformational changes are thought to polarize key residues at the start of a chain of conserved acidic and basic residues extending into and along the membrane arm, thus creating an electrostatic pulse (*Euro et al., 2008*) driving the proton pumping modules of complex I . To gain insight into the mode of ubiquinone binding and the conformation of its binding site during catalysis, we used cryo-EM to analyse the structure of mitochondrial complex I  from the aerobic yeast *Y. lipolytica* captured during steady-state turnover and in the deactive form.

## Results

### Cryo-EM structure of *Y. lipolytica* complex I  in the deactive state

A cryo-EM map of complex I  purified from *Y. lipolytica* was obtained from 124,626 particle images and refined to 4.3 Å overall resolution (*Figures 1–3*, *Table 1*). Central parts of the molecule were significantly better resolved (*Figures 2* and *3*). The final model of 42 subunits was 88% complete and contained 7515 fitted residues (*Figure 4*, *Table 2*). The cryo-EM structure of complex I  agrees well (r.m.s.d. of 1.768 Å for central subunits) with the previously determined X-ray structure in the deactive state, which was however much less complete with only 4979 residues fitted (*Figure 5*). Whereas cryo-EM of mammalian complex I  resolved several different conformations (*Zhu et al., 2016*; *Fiedorczuk et al., 2016*), we observed only one major class, indicating that the preparation of *Y. lipolytica* complex I  was homogeneous and in a uniform state (*Figure 1*).

### The central subunits

The overall structure of the fourteen central subunits (*Figure 4*) is conserved in all known complex I structures. In mammalian and *Y. lipolytica* complex I  the 49 kDa subunit harbors a long N-terminal extension that runs on the matrix side to approximately the middle of the membrane arm. In *Y. lipolytica*, an N-terminal extension of the 30 kDa subunit reaches towards the attachment site of the accessory sulfur transferase subunit ST1 (*Figure 6*). A C-terminal sequence stretch of membrane-bound subunit ND3 extends vertically along the matrix arm, forming an elongated contact site with accessory subunit NUFM (*Figure 6*). Compared with the X-ray structure of *Y. lipolytica* complex I (*Zickermann et al., 2015*) we changed the assignment of TMH4 of subunit ND6. The corresponding helices match the X-ray structure of *Thermus thermophilus* and the cryo-EM structure of *Y. lipolytica* complex I  but are shifted towards the peripheral arm in mammalian complex I  (*Figure 7*). In conjunction with the absence of TMH1-3 of ND2, this causes a conspicuous ambilateral indentation of the membrane arm of mammalian complex I  that is not present in bacterial and yeast complex I .

### The accessory subunits

We identified density for all expected 28 accessory subunits of *Y. lipolytica* complex I  (*Figure 8*, *Table 2*). In the membrane arm we detected density for a new accessory subunit with a single transmembrane helix (provisionally labelled X in *Figure 8*). With the exception of this previously unknown subunit and subunits ST1 and NUXM, all accessory subunits of *Y. lipolytica* complex I were assigned to corresponding subunits of mammalian complex I  based on sequence alignments and structural correlation (*Table 2*). For eighteen accessory subunits the cryo-EM map showed most or almost all (59–98%) of the side chain densities (*Figure 8—figure supplement 1*). An overlay of eight subunits modelled completely or mainly as poly-alanine on their bovine counterparts is shown in *Figure 8—figure supplement 2*. Subunits NESM and NUZM have prominent extra domains and the NUFM subunit carries an N-terminal extension that interacts tightly with the C-terminal extension of central subunit ND3 (*Figure 6*). In contrast, NI2M, NB5M and NIDM of *Y. lipolytica* are significantly smaller.

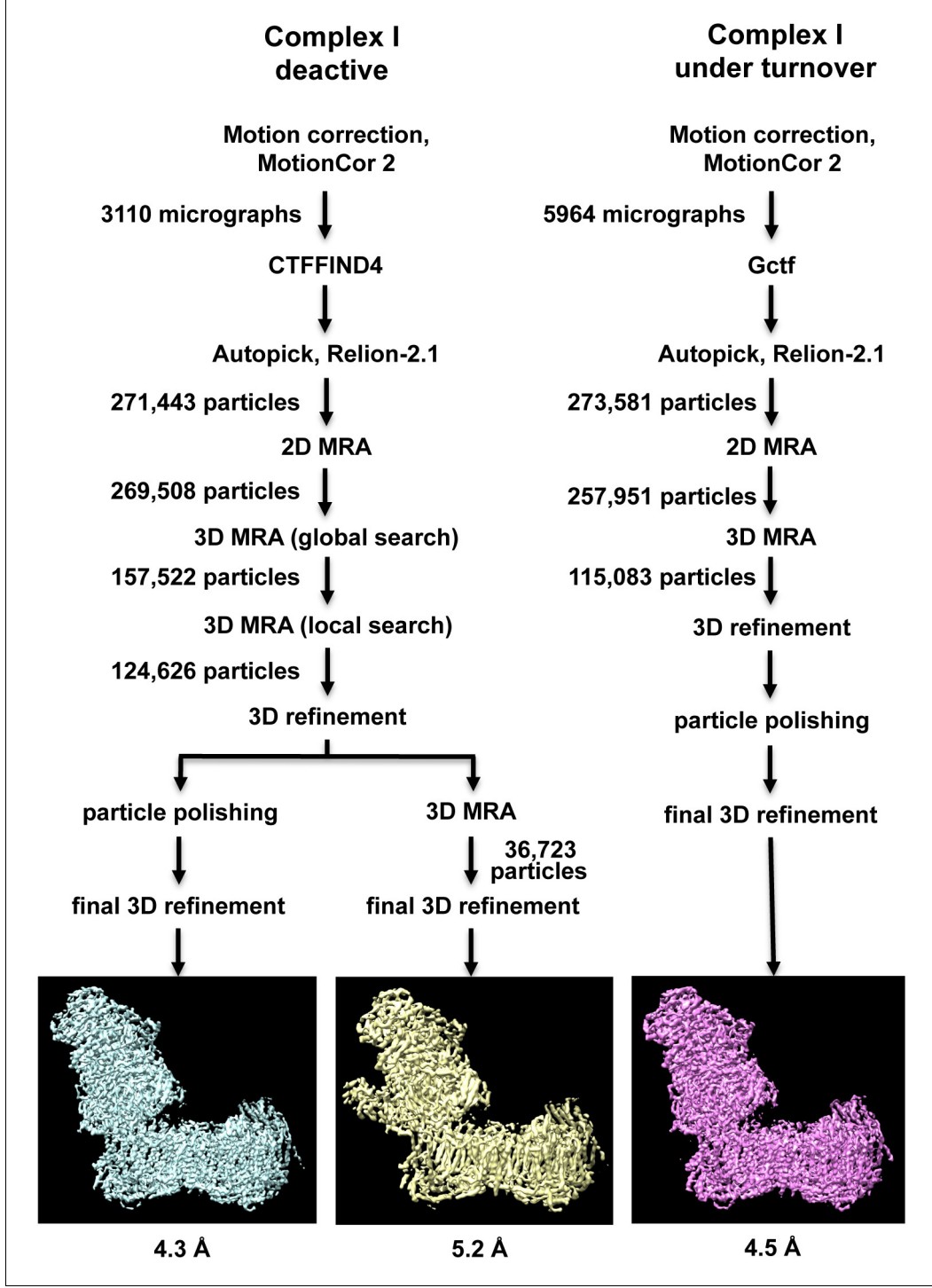

**Figure 1.** Image processing and two-dimensional classification of particle images. Electron densities of complex I in the deactive state and under turnover conditions are light blue and magenta. About 30% of the particles belong to a subclass (yellow) which contains the accessory sulfur transferase subunit ST1 known to be bound substoichiometrically to *Y. lipolytica* complex I (***D'Imprima et al., 2016***).

DOI: https://doi.org/10.7554/eLife.39213.002

The following figure supplement is available for figure 1:

**Figure supplement 1.** Raw micrographs and class averages.

DOI: https://doi.org/10.7554/eLife.39213.003

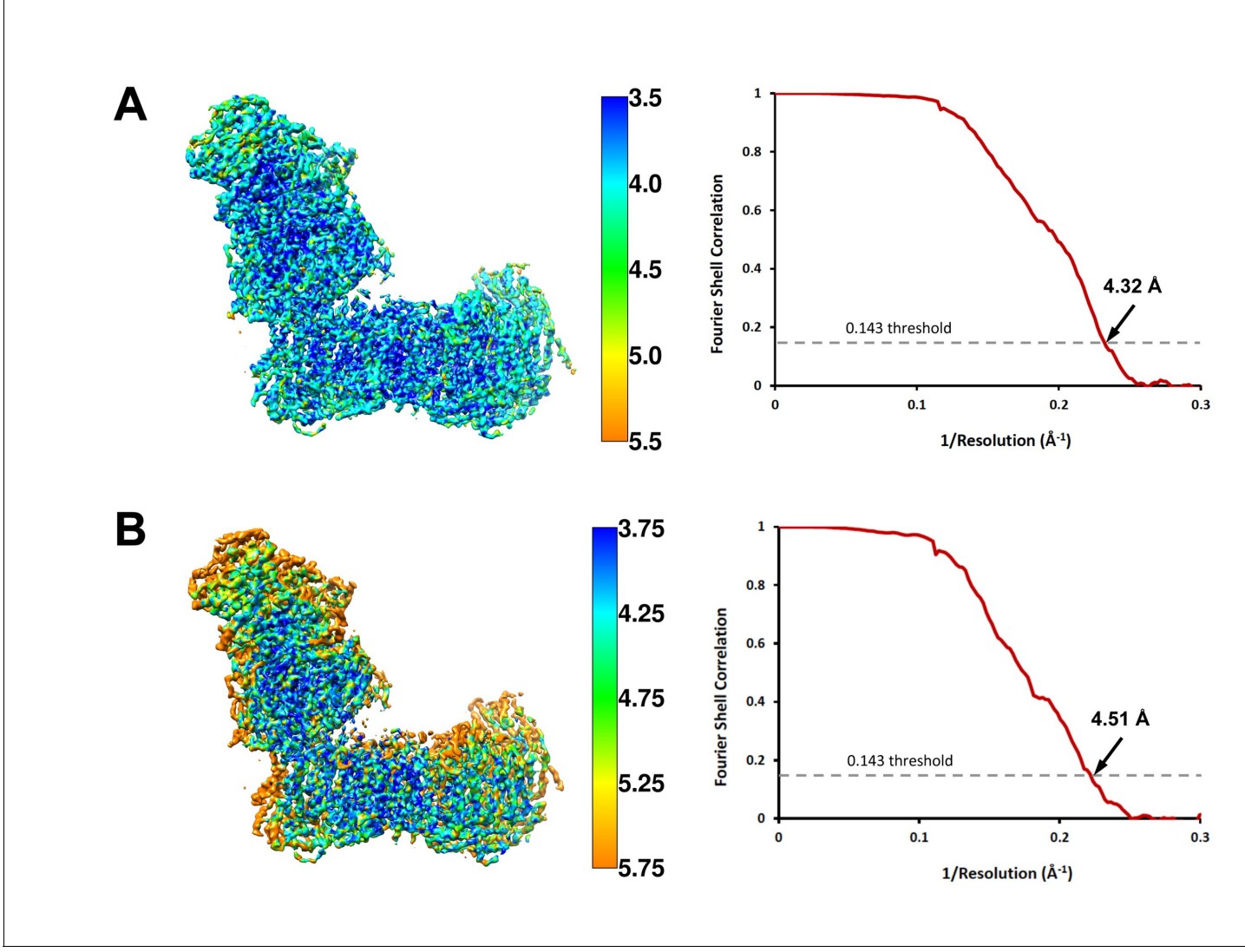

**Figure 2.** Local resolution and Fourier shell correlation (FSC) curves of (**A**) deactive complex I and (**B**) complex I under steady-state turnover conditions. Left: cryo-EM maps of complex I analysed by ResMap (*Kucukelbir et al., 2014*) coloured according to local resolution. Right: FSC plots of final masked and refined cryo-EM maps. The map resolution is indicated by the point where the curve crosses the 0.143 threshold (*Rosenthal and Henderson, 2003*).

DOI: https://doi.org/10.7554/eLife.39213.004

We did not find any density at positions of the mammalian 42 kDa subunit (NDUFA10), the 9 kDa subunit (NDUFV3), or the MNLL (NDUFB1) subunit, consistent with the absence of their orthologs in the *Y. lipolytica* genome. The absence of the MNLL subunit correlates with a shift of the adjacent TMH1 in ND4 (*Figure 7*). We modelled subunit NUXM that has no correspondence in vertebrates next to the first three helices of central subunit ND2, which are absent in mammalian complex I (*Figure 8—figure supplement 1*). This assignment is based on the allocation of NUXM to the $P_P$ module (*Angerer et al., 2011*), secondary structure prediction, and a short stretch of sequence with side chains. The sulfur transferase subunit ST1 was present sub-stoichiometrically as observed before (*Figure 1*) (*D'Imprima et al., 2016*). Based on biochemical data we had suggested that subunit N7BM anchors ST1 to the peripheral arm (*Kmita et al., 2015*). We now show that N7BM shares an extensive interface with ST1, and that ST1 furthermore interacts with NUZM and the N-terminal region of the 30 kDa subunit (*Figure 6*).

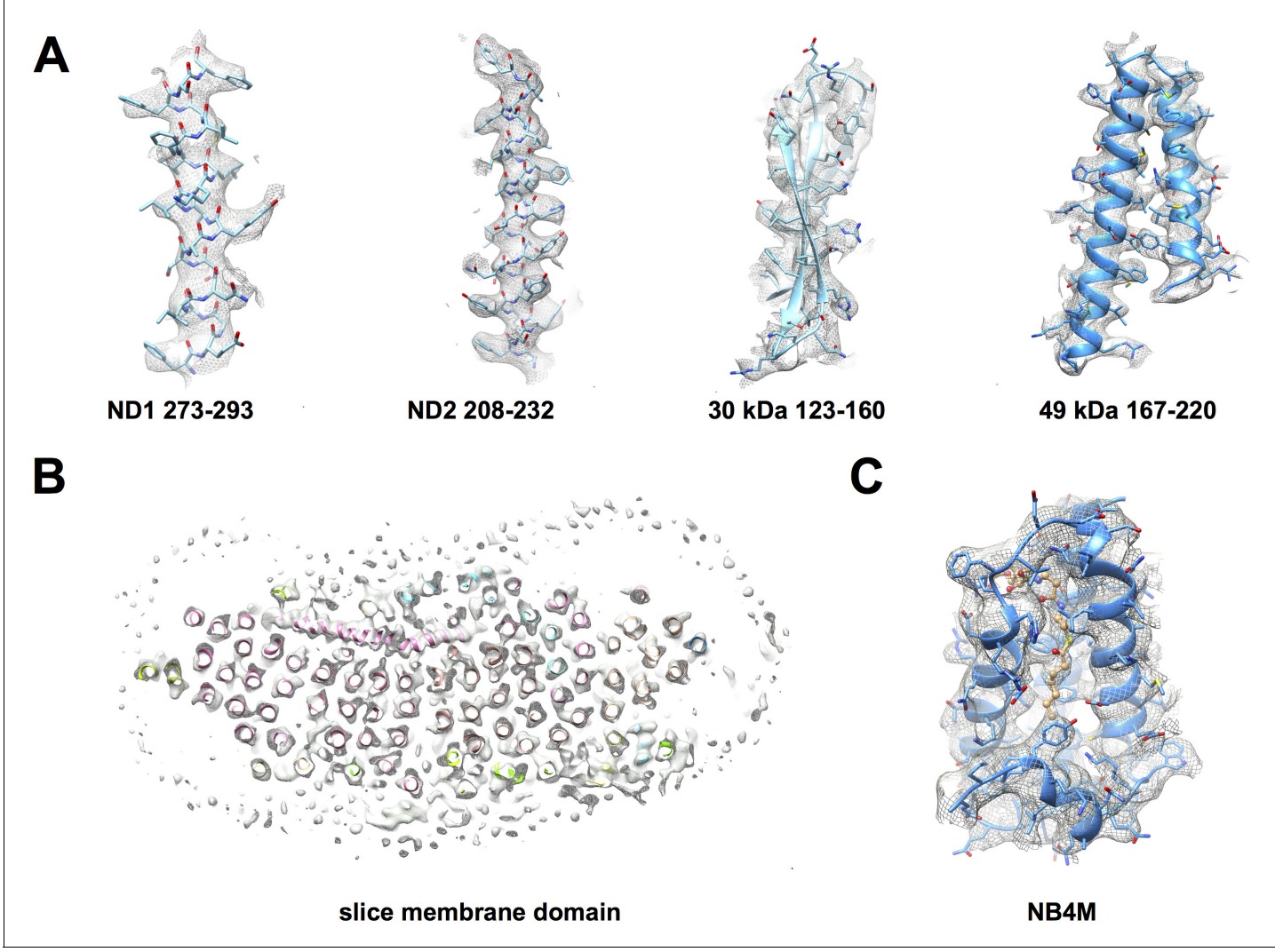

**Figure 3.** Cryo-EM map of deactive complex I with fitted models. (A) Selected regions of the matrix domain. (B) A horizontal cross-section through the membrane arm shows TMH fits. (C) A region of the accessory LYR protein subunit NB4M (*Angerer et al., 2017*); the acyl chain appended to the phosphopantetheine group of the adjacent acyl carrier protein ACPM1 inserts into the interior of NB4M. Cofactor and acyl chain are drawn as ball-and-stick model.

DOI: https://doi.org/10.7554/eLife.39213.005

The following figure supplement is available for figure 3:

**Figure supplement 1.** Validation of model refinement.

DOI: https://doi.org/10.7554/eLife.39213.006

## Lipid molecules and an unassigned density in the interface region

Four lipid molecules were identified in the membrane arm. A prominent density in the interior of subunit ND1 towards the interface with the PSST and 49 kDa subunits was not assigned unambiguously (*Figure 9*). The most prominent part of this density was in direct contact with side chains of conserved Arg36$^{ND1}$, Arg297$^{ND1}$ and Leu200$^{49-kDa}$, and close to the conserved Arg27$^{ND1}$ and Arg108$^{PSST}$. The density is remarkable because it is close to or even within the proposed access pathway by which ubiquinone enters the active site. Recent MD simulations suggested that the residues corresponding to Arg27$^{ND1}$ and Arg108$^{PSST}$ in bovine complex I would coordinate part of the isoprenoid side chain of ubiquinone by π-stacking of their guanidinium groups (*Fedor et al., 2017*). In contrast, the narrow and well-defined contact sites with Arg36$^{ND1}$ and Arg297$^{ND1}$ we observe are consistent with a strong ionic interaction. The density appears to be too bulky for an isoprenoid

**Table 1.** Data collection, refinement and model statistics.

|  | **Deactive** | **steady-state turnover** |
| --- | --- | --- |
| Data collection<br>Microscope | FEI Tecnai Polara | FEI Titan Krios |
| Camera | Gatan K2 Summit | Falcon III |
| Voltage (kV)<br>Nominal magnification<br>Calibrated pixel size (Å) | 300<br>200,000x<br>1.09 | 300<br>75,000x<br>1.053 |
| Total exposure ($e^-/Å^2$) | 60.5 | 30.7–40.7 |
| Exposure rate ($e^-$/pixel/s)<br>Number of frames | 9<br>40 | 0.4<br>81 |
| Defocus range (μm) | 1.5–3.0 | 1.5–4.5 |
| Image processing<br>Motion correction software | *MotionCor2* | *MotionCor2* |
| CTF estimation software | *CTFFIND4* | *Gctf* |
| Particle selection software | *EMAN* boxer and *RELION2.1* | *RELION2.1* |
| Initial/final micrographs | 3,110/3,110 | 5,964/5,650 |
| Particles selected | 271,443 | 273,581 |
| Applied *B*-factor ($Å^2$) | −142 | −215 |
| Final resolution (Å) | 4.3 | 4.5 |
| Refinement statistics<br>Modeling software | *COOT, PHENIX* | |
| Number of residues | 7515 | |
| Map CC (whole unit cell) | 0.786 | |
| RMS deviations<br>Bond-lengths (Å) | 0.0099 | |
| Bond-angles (°) | 1.52 | |
| Av. B-factor ($Å^2$) | 151.46 | |
| Ramachandran plot<br>Outliers (%) | 0.74 | |
| Allowed (%) | 13.43 | |
| Favoured (%) | 85.83 | |
| Rotamer outliers (%) | 0.71 | |
| Molprobity score | 2.11 | |
| All-atom clashscore | 8.55 | |
| PDB ID | 6GCS | |

DOI: https://doi.org/10.7554/eLife.39213.009

chain or even for a ubiquinone head group. We therefore suggest that the density represents the negatively charged head group of a lipid molecule. More than 60 lipid molecules per complex I were bound in our preparation as detected by mass spectrometry (*Table 3*), but at our current map resolution their densities cannot be assigned unequivocally. A lipid head group would be consistent with the complementary charges of conserved arginine residues in this area, although it would be situated at an unusual position above the membrane surface. The same density was found under turnover conditions (see below) and therefore obviously did not prevent the passage of ubiquinone to its binding site. Although a lipid molecule in this position might partly block the generally accepted ubiquinone access pathway through the portal formed by TMH1, TMH6, and helix α1–2 of ND1, it would be flexible and mobile enough to allow ubiquinone access, while keeping other hydrophilic compounds out, acting like a hydrophobic valve.

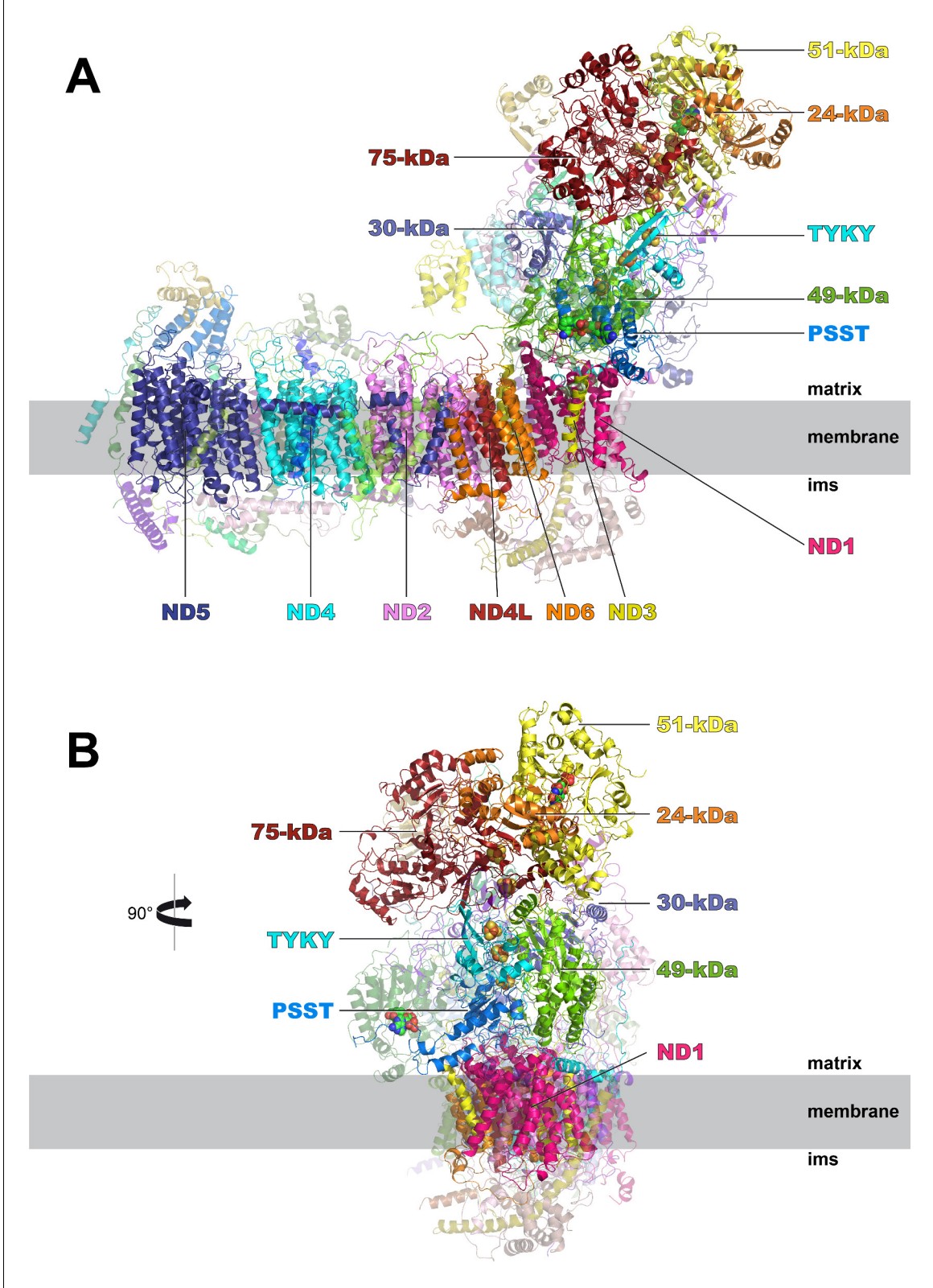

**Figure 4.** Cryo-EM structure of respiratory complex I from *Y. lipolytica*. (**A**) Side view; (**B**) view from peripheral arm; central subunits (labelled, solid) and accessory subunits (transparent, compare *Figure 8*) are shown.

DOI: https://doi.org/10.7554/eLife.39213.007

**Table 2.** Composition of subunits.

| Subunit | human/bovine | Chain | Total residues/ range built | Modelled with side chains | Modelled as poly-alanine | [%] residues modelled | [%] with side chains | [%] unknown |
|---|---|---|---|---|---|---|---|---|
| central subunits | | | | | | | | |
| NUAM | NDUFS1/ 75 kDa | A | 694/ 1–691 | 1–691 | 0 | 99 | 99 | 1 |
| NUBM | NDUFV1/ 51 kDa | B | 470/ 15–457 | 15–437 | 437–457 | 95 | 90 | 6 |
| NUCM | NDUFS2/ 49 kDa | C | 444/ 28–443 | 58–67 77–443 | 28–57 68–76 | 93 | 85 | 7 |
| NUGM | NDUFS3/ 30 kDa | G | 251/ 1–232 | 30–189 | 1–29 190–232 | 92 | 64 | 8 |
| NUHM | NDUFV2/ 24 kDa | H | 215/ 3–187 | 23–187 | 3–22 | 86 | 78 | 14 |
| NUIM | NDUFS8/ TYKY | I | 198/ 19–198 | 26–198 | 19–26 | 91 | 87 | 9 |
| NUKM | NDUFS7/ PSST | K | 183/ 15–183 | 15–183 | 0 | 92 | 92 | 8 |
| NU1M | NU1M/ ND1 | 1 | 341/ 1–340 | 1–179 184–205 217–251 268–340 | 180–183 206–216 252–267 | 100 | 91 | 0 |
| NU2M | NU2M/ ND2 | 2 | 469/ 1–85 99–465 | 1–25 53–85 99–415 | 26–52 416–465 | 96 | 80 | 4 |
| NU3M | NU3M/ ND3 | 3 | 128/ 1–34 49–124 | 1–34 49–118 | 119–124 | 81 | 77 | 19 |
| NU4M | NU4M/ ND4 | 4 | 486/ 7–481 | 85–189 201–434 | 7–84 190–200 435–481 | 98 | 70 | 2 |
| NU5M | NU5M/ ND5 | 5 | 655 5–479 489–652 | 28–436 457–474 568–592 614–652 | 5–27 437–456 475–479 489–567 593–613 | 98 | 75 | 2 |
| NU6M | NU6M/ ND6 | 6 | 185 2–184 | 2–77 160–184 | 78–159 | 99 | 55 | 1 |
| NULM | NULM/ ND4L | L | 89 1–86 | 1–86 | 0 | 97 | 97 | 3 |
| accessory subunits peripheral arm | | | | | | | | |
| NUEM | NDUFA9/ 39 kDa | E | 355 17–334 | 17–334 | 0 | 82 | 82 | 18 |
| NUFM | NDUFA5/ B13 | F | 136 13–131 | 34–131 | 13–33 | 88 | 72 | 12 |
| NUMM | NDUFS6/ 13 kDa | M | 119 13–117 | 43–117 | 13–42 | 88 | 75 | 12 |
| NUYM | NDUFS4/ 18 kDa | Y | 137 19–133 | 19–133 | 0 | 84 | 84 | 16 |
| NUZM | NDUFA7/ B14.5a | Z | 182 30–166 | 0 | 30–166 | 75 | 0 | 25 |
| N7BM | NDUFA12/B17.2 | h | 137 6–135 | 6–135 | 0 | 95 | 95 | 5 |
| NB4M | NDUFA6/ B14 | P | 123 3–118 | 3–118 | 0 | 94 | 94 | 6 |
| ACPM1 | NDUFAB1/SDAP | O | 84 4–80 | 4–80 | 0 | 92 | 92 | 8 |

*Table 2 continued on next page*

*Table 2 continued*

| Subunit | human/bovine | Chain | Total residues/ range built | Modelled with side chains | Modelled as poly-alanine | [%] residues modelled | [%] with side chains | [%] unknown |
|---|---|---|---|---|---|---|---|---|
| NI8M | NDUFA2/ B8 | f | 86 5–84 | 5–84 | 0 | 93 | 93 | 7 |
| Accessory subunits P$_P$ module | | | | | | | | |
| NUPM | Ndufa8/ pgiv | U | 171 10–167 | 16–142 | 10–15 143–167 | 92 | 74 | 8 |
| NUJM | NDUFA11/B14.7 | J | 197 18–157 | 18–133 | 134–157 | 71 | 59 | 29 |
| NB6M | NDUFA13/B16.6 | W | 122 3–120 | 14–120 | 3–13 | 97 | 88 | 3 |
| NIPM | NDUFS5/ 15 kDa | 9 | 88 8–66 | 14–66 | 08–13 | 67 | 60 | 33 |
| NUXM | -/- | X | 168 4–120 | 72–96 | 4–71 97–120 | 70 | 15 | 30 |
| NI9M | NDUFA3/ B9 | g | 66 3–62 | 3–62 | | 91 | 91 | 9 |
| NIMM | NDUFA1/ MWFE | D | 86 1–80 | 1–58 | 59–80 | 93 | 67 | 7 |
| NEBM | NDUFC2/ B14.5b | b | 73 1–64 | | 1–64 | 87 | 0 | 13 |
| accessory subunits P$_D$ module | | | | | | | | |
| NESM | NDUFB11/ESSS | S | 204 29–187 | | 29–187 | 78 | 0 | 22 |
| NIAM | NDUFB8/ ASHI | a | 125 11–110 | | 11–110 | 80 | 0 | 20 |
| NUNM | NDUFB5/ SGDH | n | 119 23–115 | | 23–115 | 78 | 0 | 22 |
| NB2M | NDUFB3/ B12 | c | 59 8–49 | 31–49 | 8–30 | 71 | 32 | 29 |
| NB5M | NDUFB4/ B15 | j | 92 3–75 | 16–41 | 3–15 42–75 | 82 | 27 | 18 |
| NB8M | NDUFB7/ B18 | 8 | 98 3–84 | 3–80 | 81–84 | 84 | 80 | 16 |
| ACPM2 | NDUFAB1/SDAP | Q | 88 3–87 | 3–87 | 0 | 97 | 97 | 3 |
| NIDM | NDUFB10/PDSW | d | 91 3–91 | 3–91 | 0 | 98 | 98 | 5 |
| NI2M | NDUFB9/ B22 | R | 108 6–107 | 6–98 | 99–107 | 94 | 86 | 6 |
| NUUM | NDUFB2/ AGGG | e | 89 6–50 | | 6–50 | 51 | 0 | 49 |

DOI: https://doi.org/10.7554/eLife.39213.010

## The ubiquinone binding site in the deactive state

The ubiquinone binding and access site is essentially formed by the hydrophilic PSST and 49 kDa subunits and the membrane-intrinsic ND1 subunit (*Zickermann et al., 2015*). In recent cryo-EM structures of deactive mammalian complex I , critical loops of the latter two subunits and of the adjacent ND3 and the accessory 39 kDa subunits were proposed to unfold, because no matching density was found (*Blaza et al., 2018*; *Agip et al., 2018*). In our map only residues 35 – 48 in the central part of the long TMH1-2 loop of subunit ND3 were disordered (*Figure 10*). In contrast to bovine complex I , we observed continuous density for the loop connecting the first two β-strands of the 49 kDa subunit and for the TMH5-6 loop of ND1. The local resolution of this region indicated some degree of flexibility, and it was therefore modelled as poly-alanine. The C-terminal domain of

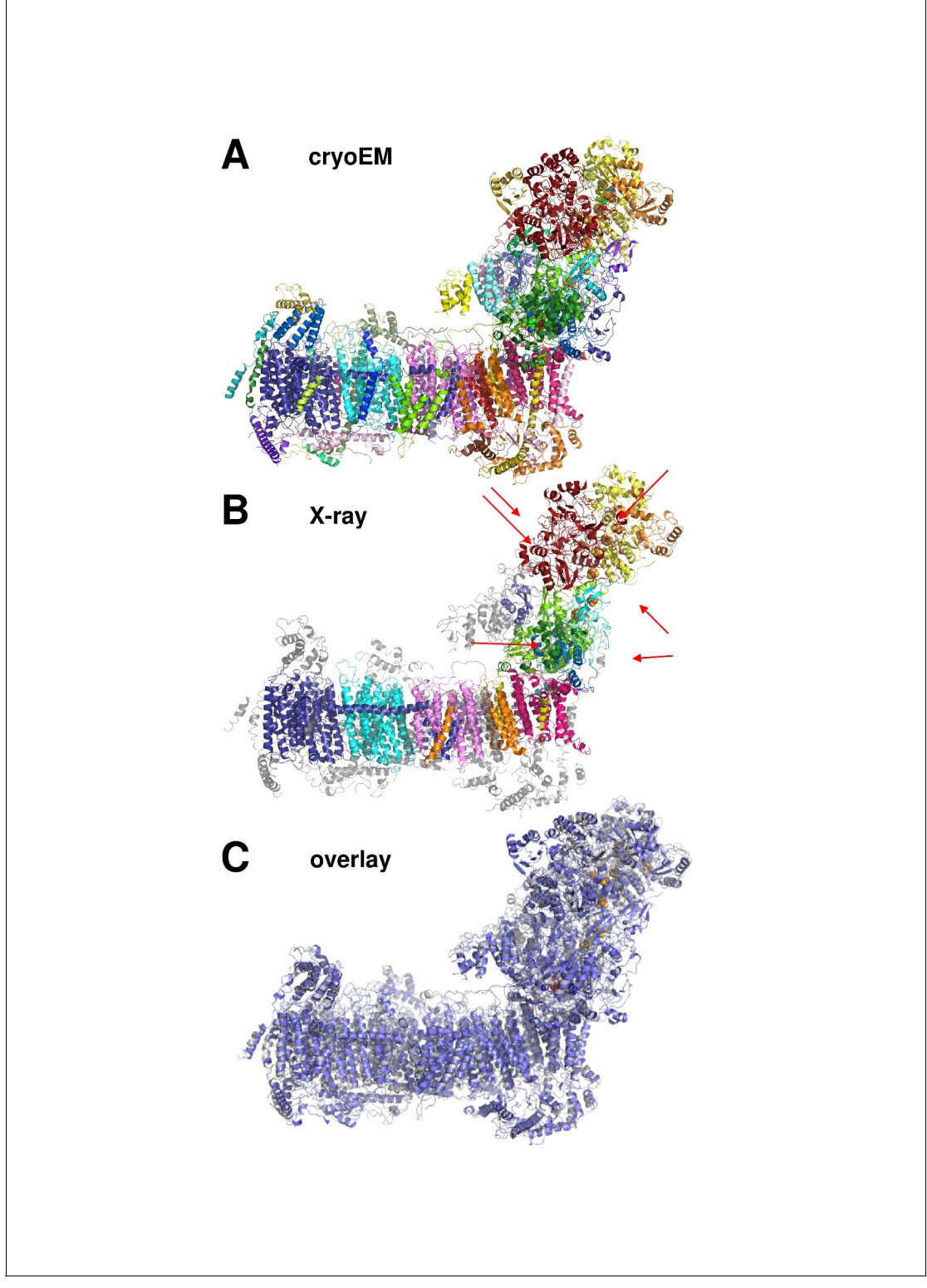

**Figure 5.** Cryo-EM and X-ray structures of deactive complex I are consistent. (**A**) With 42 assigned subunits (colour coded as in *Figures 4* and *8*) and 7515 residues the cryo-EM structure is significantly more complete than the X-ray structure (**B**) with 15 assigned subunits and 4979 residues (colour coded as in (**A**). Unassigned parts of the model are grey. Red arrows indicate subunits NI8M, NUYM, NUZM, N7BM, NUMM, cofactors FMN (51 kDa subunit) and NADPH (NUEM subunit) that are missing or incomplete in the X-ray structure. (**C**) X-ray structure of deactive complex I from *Y. lipolytica* (*Zickermann et al., 2015*) (grey) overlaid with the cryo-EM structure of deactive *Y. lipolytica* complex I (blue).

DOI: https://doi.org/10.7554/eLife.39213.008

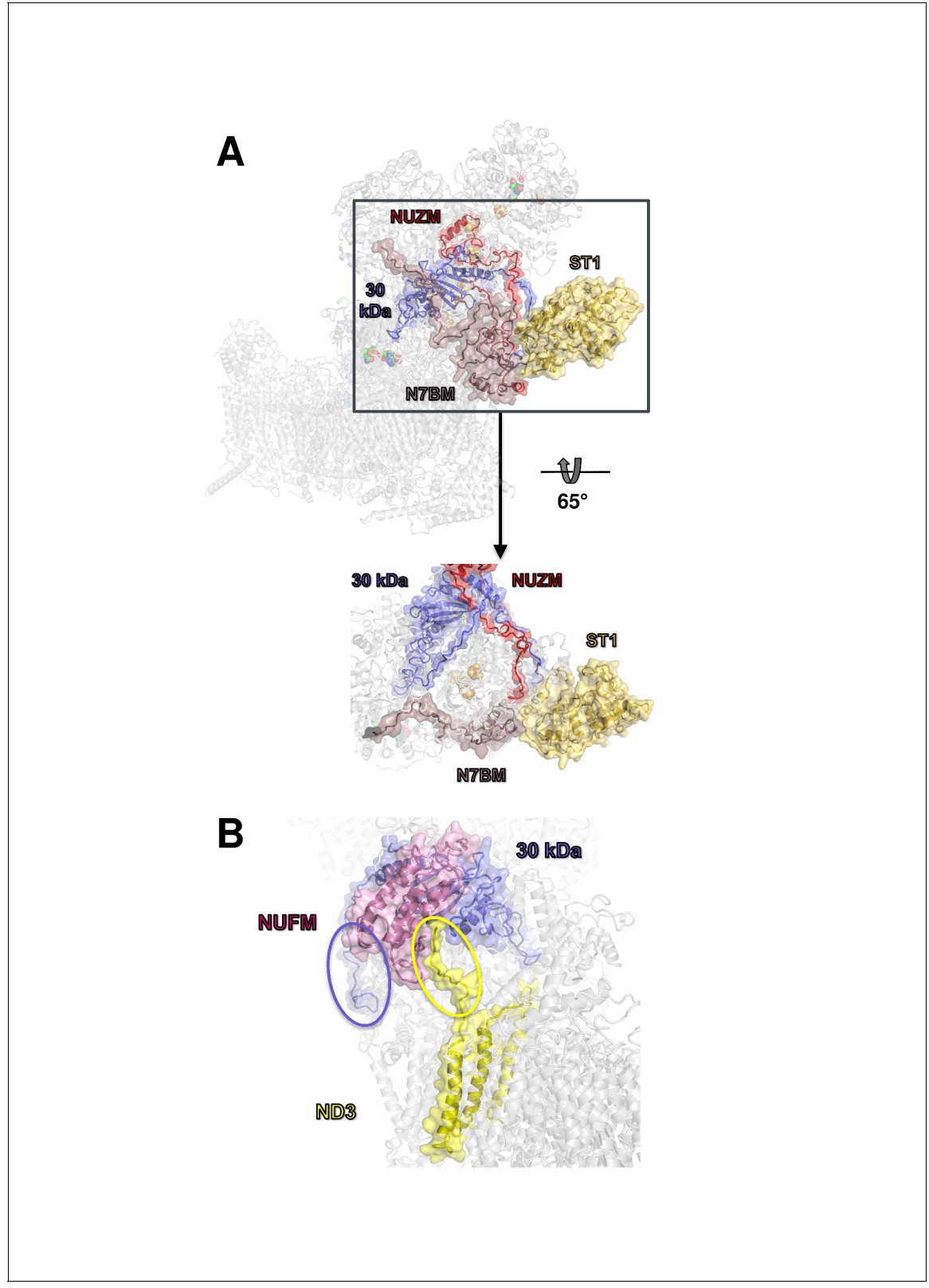

**Figure 6.** Docking site of accessory subunit ST1 and extensions of the 30 kDa and ND3 subunits in *Y. lipolytica* complex I . (**A**) ST1 (yellow) binds to N7BM (violet), NUZM (red) and the extended N-terminus of the 30 kDa subunit (blue); (**B**) N-terminal extension of the 30 kDa subunit (blue oval) and interaction of the C-terminal extension of subunit ND3 with NUFM (yellow oval).

DOI: https://doi.org/10.7554/eLife.39213.011

accessory subunit NUEM, which is supposed to have a regulatory function in the A/D transition (*Babot et al., 2014*), was largely modelled with side chains, whereas the corresponding domain in the mammalian 39 kDa subunit was disordered.

## Cryo-EM structure under steady-state turnover conditions

To analyse substrate binding and structural changes during the catalytic cycle we collected cryo-EM images of complex I under turnover conditions, after adding NADH and ubiquinone to the sample. Ubiquinone is hydrophobic and even the short-chain analogue decylubiquinone (DBQ) used in this study is not soluble in aqueous buffers at sufficiently high concentrations. To avoid substrate exhaustion at 2 µM complex I concentration between adding the substrate and freezing the sample, we reduced the temperature of the reaction to 18°C and added 1 µM ubiquinol oxidase from *Vitreoscilla* sp. to recycle the ubiquinone. An increase in oxidase concentration did not increase the rate of oxygen consumption, indicating that complex I was rate-limiting and that ubiquinone oxidation was efficient. Recycling of substrate by this artificial respiratory chain sustained steady-state conditions for more than one minute, while freezing the cryo-EM sample took only 20–30 s (*Figure 11*).

The structure of complex I under turnover conditions indicated no overall changes compared to the deactive state. We can therefore exclude large conformational rearrangements, such as the proposed extensive movement of the two arms relative to one another (*Böttcher et al., 2002*) (*Figure 12*). We also found no evidence for a piston-like movement of the long lateral helix of the membrane arm that has been proposed to couple redox chemistry to proton translocation (*Hunte et al., 2010*; *Baradaran et al., 2013*). The most obvious difference was a strong additional density in the 51 kDa subunit, which was modelled as bound NADH (*Figure 13A*) in agreement with a previous structure of bacterial complex I (*Berrisford and Sazanov, 2009*). A slight movement of

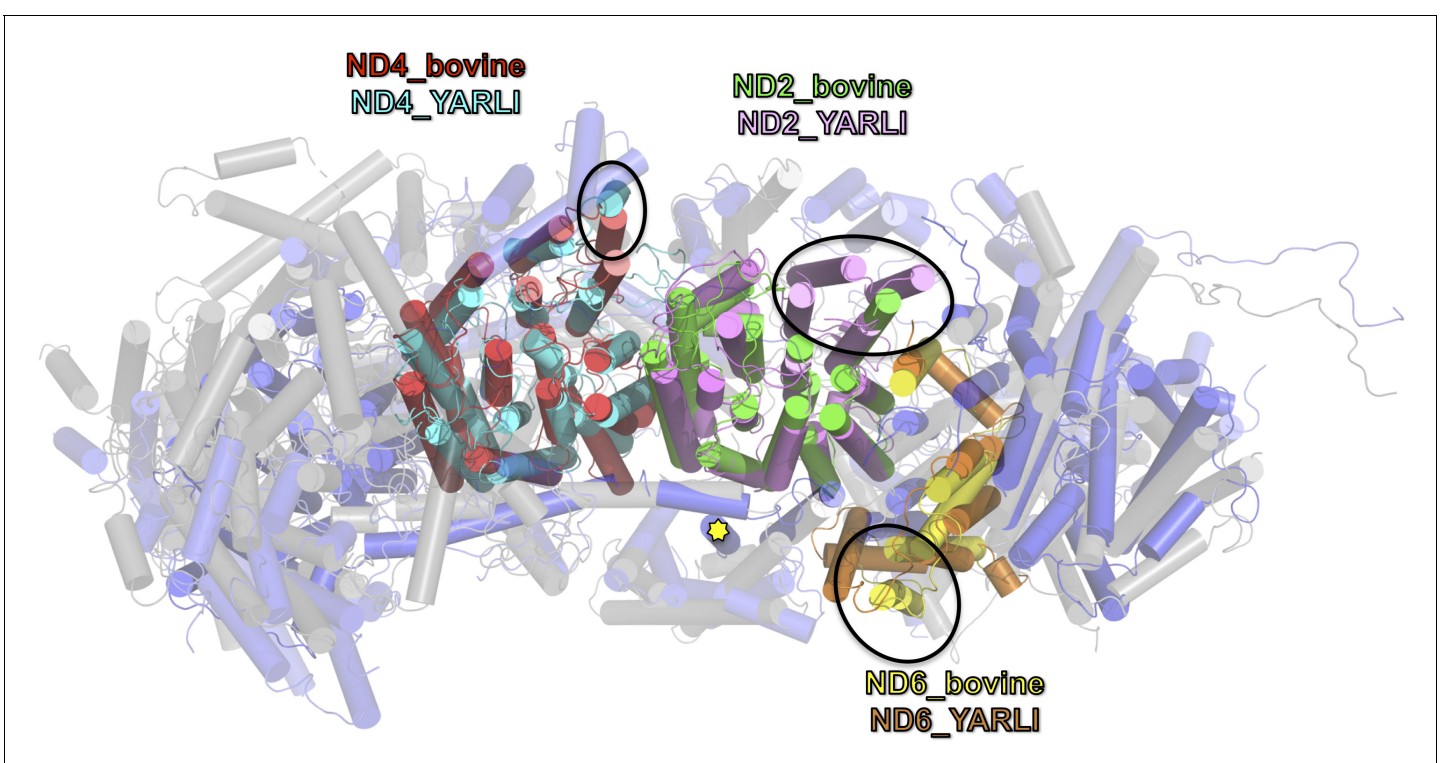

**Figure 7.** Overlay of membrane arm subunits of complex I from *Y. lipolytica* and *B. taurus*. Top view of membrane arm with subunits of the peripheral arm removed for clarity (*Y. lipolytica* blue, *B. taurus*, grey; selected subunits are coloured as indicated). The first three helices of ND2 are missing in bovine complex I and the position of TMH 4 of ND6 is different. These changes result in an incision of the membrane arm of mammalian complex I at the position of ND2. Subunit ND5 of complex I from *Y. lipolytica* has an extra C-terminal TMH (yellow asterisk [*Zickermann et al., 2015*]), and TMH 1 of ND4 is oriented differently in the membrane.

DOI: https://doi.org/10.7554/eLife.39213.012

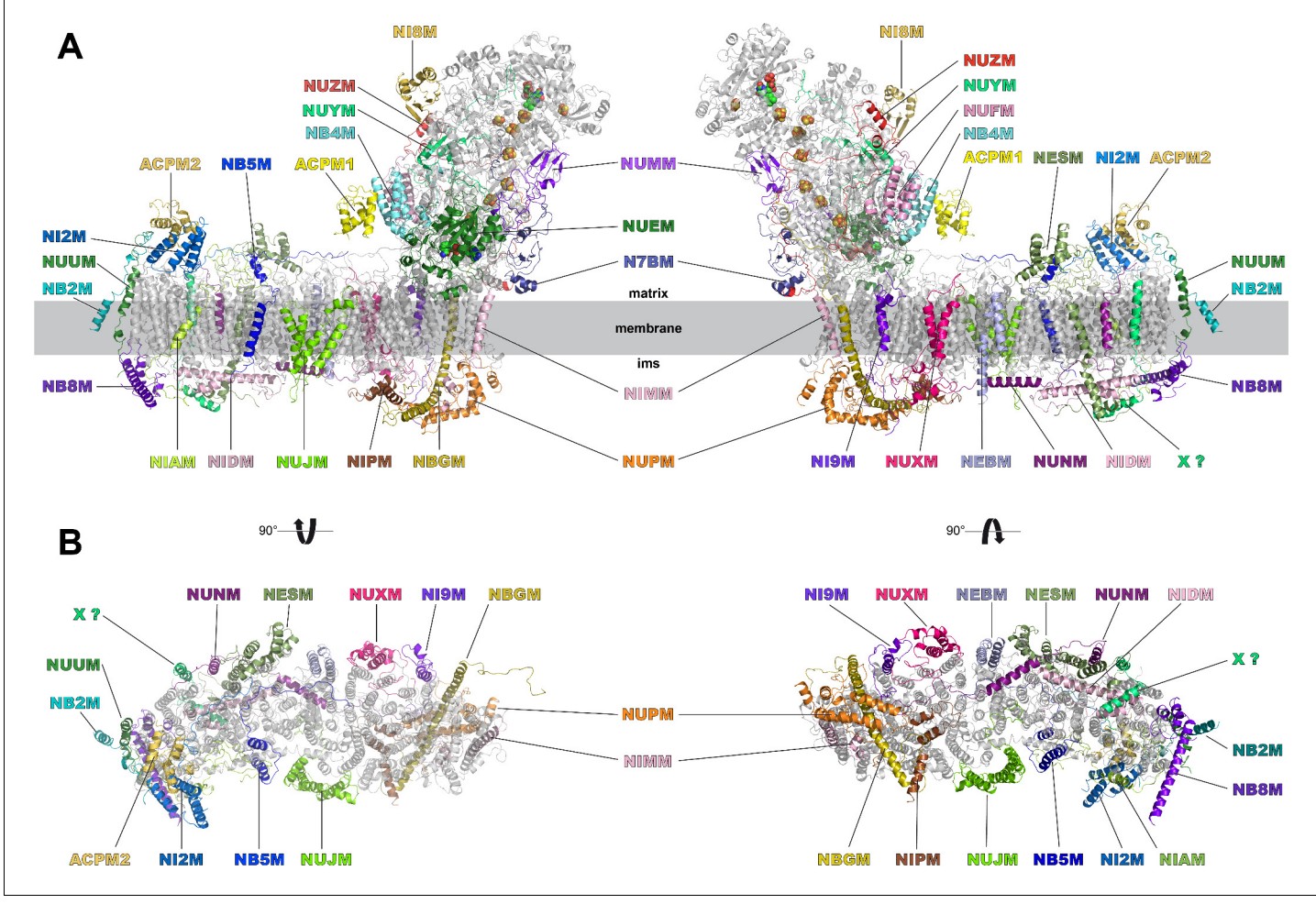

**Figure 8.** Accessory subunits of complex I . Central subunits (see *Figure 4*) are shown in grey, accessory subunits are labelled and coloured. (A) Side views, (B) view from the matrix (left) and from the intermembrane space (right) with peripheral arm subunits removed for clarity.

DOI: https://doi.org/10.7554/eLife.39213.013

The following figure supplements are available for figure 8:

**Figure supplement 1.** Assignment of accessory subunits.

DOI: https://doi.org/10.7554/eLife.39213.014

**Figure supplement 2.** Assignment of accessory subunits.

DOI: https://doi.org/10.7554/eLife.39213.015

the glycine-rich loop (Gly88-Gly91) opens the site to accommodate the substrate in this position. The pyridine ring moiety engages in a tight stacking interaction with the isoalloxazine ring of FMN to allow efficient hydride transfer. The adenine ring interacts by π-stacking with Phe92, Phe100, and Phe231 of the 51 kDa subunit. Lys228 forms a salt bridge with the pyrophosphate, and Ser325 establishes a hydrogen bond to a ribose group of the bound nucleotide.

In the 49 kDa subunit we observed a change in the structure of the β1-β2 loop of the N-terminal β sheet, which widens the central cavity formed by this subunit and subunit PSST. A clear density consistent with a ubiquinone head group was found between the 49 kDa subunit β1-β2 loop and the α2 helix of subunit PSST (*Figure 13B,C*). This position closely matches that of a ubiquinone-antagonistic inhibitor in the X-ray structure of *Y. lipolytica* complex I , as identified by anomalous scattering of bromine-substituted inhibitor (*Figure 13D*) (*Zickermann et al., 2015*). We conclude that the new density represents the headgroup of a bound ubiquinone substrate at a minimal edge-to-edge distance of ~12 Å from cluster N2. This site is different from the ubiquinone binding site reported for the bacterial enzyme from *T. thermophilus*, which engages a strictly conserved tyrosine adjacent to

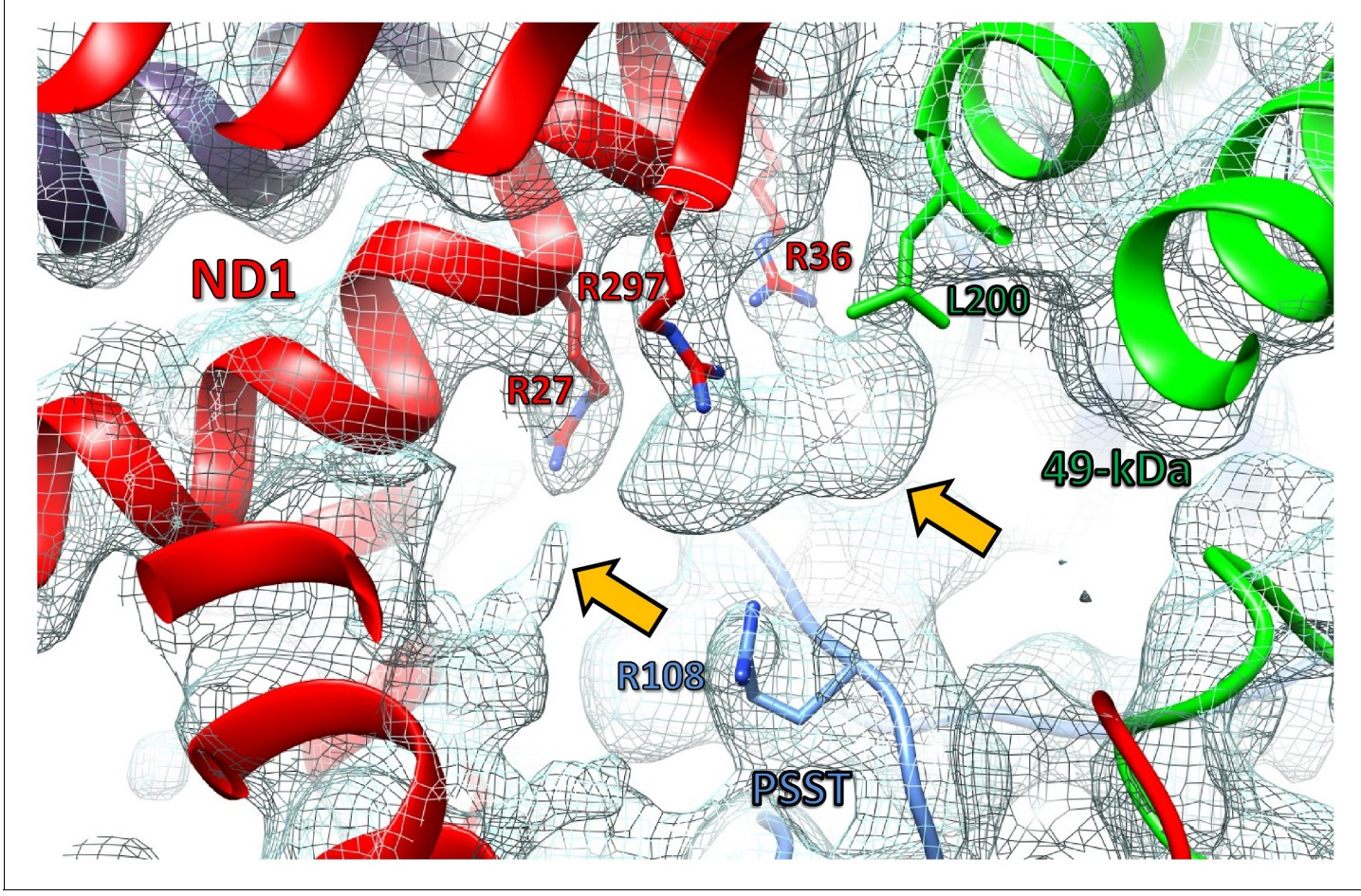

**Figure 9.** Unassigned density (orange arrows) at the interface of subunits ND1, 49 kDa, and PSST. Slice of interface region of membrane and peripheral arm of complex I in the deactive state (model, cartoon representation; selected residues, stick representation; cryo-EM map, mesh). Note that the density is also present in the maps of complex I under steady-state turnover conditions.
DOI: https://doi.org/10.7554/eLife.39213.016

the terminal iron-sulfur cluster N2 and positions the quinone headgroup ~2 Å closer to cluster N2 (*Baradaran et al., 2013*) (*Figure 14A*). There is abundant evidence from site-directed mutagenesis in *Y. lipolytica* to support substrate and inhibitor binding to this tyrosine (*Tocilescu et al., 2010*), suggesting that this binding site also exists in the mitochondrial enzyme and that it alternates with the site observed in the structure of complex I under turnover conditions, as presented here.

## Discussion

Two different models for the structural basis of the A/D transition of complex I have recently been proposed. In the X-ray structure of the deactive form of *Y. lipolytica* complex I (*Zickermann et al., 2015*), access of a ubiquinone-antagonistic inhibitor into a binding position close to cluster N2 was blocked by a specific conformation of the β1-β2 loop of the 49 kDa subunit. We hypothesized that during the A/D transition the interface region of complex I can switch between two defined conformational states and that the A- and D-forms of complex I are tightly linked with intermediates of the catalytic cycle. In contrast, the 'unfolded Q site model' for the A/D transition (*Blaza et al., 2018*) describes the D-form as an 'off pathway' intermediate that is characterized by extensive relaxation of several loops in and around the ubiquinone binding cavity. According to this model, ubiquinone is required to restructure the interface region during transition into the A-form. Blaza *et al.* speculated that a corresponding loss of defined protein structure in the deactive *Y. lipolytica* complex I was

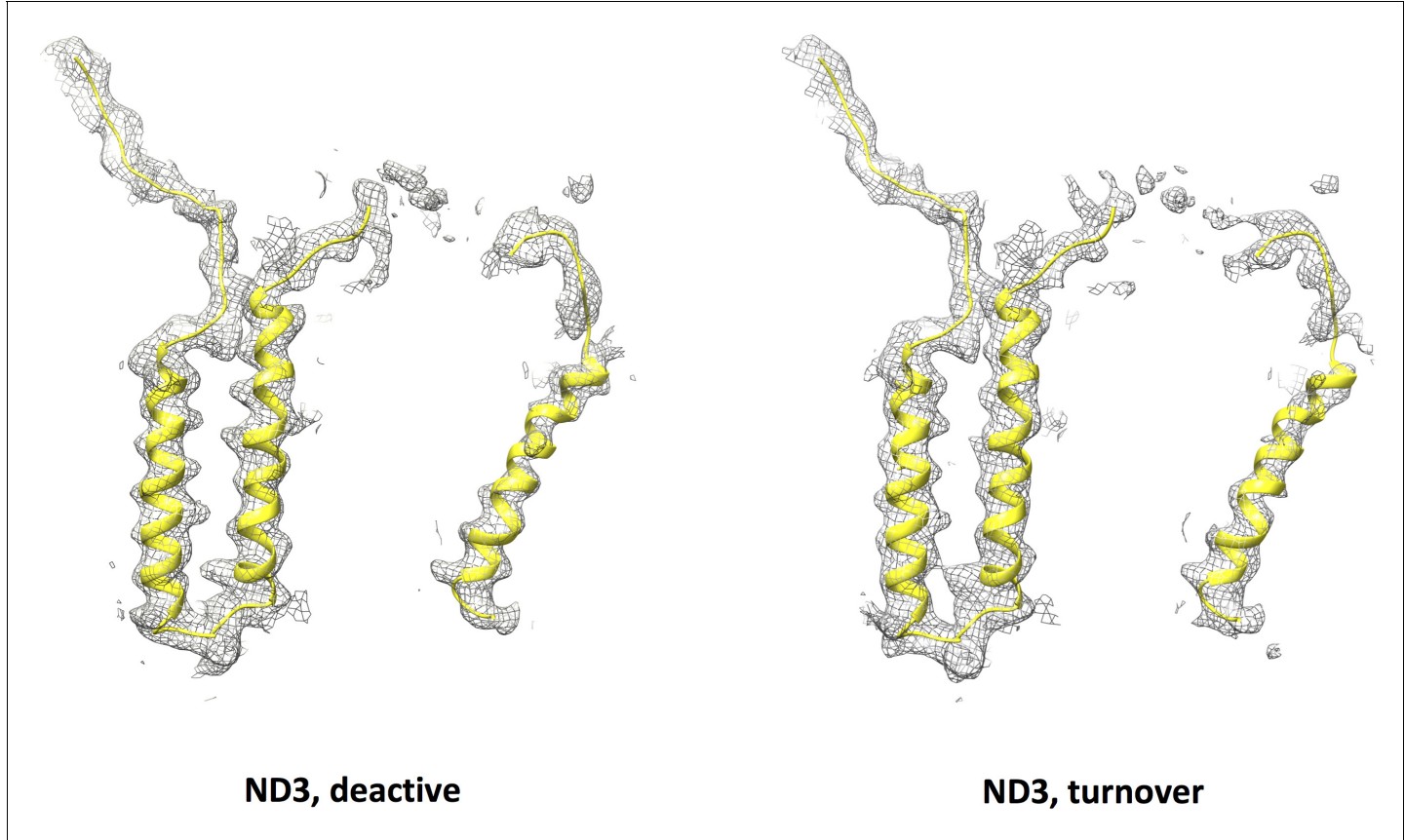

**ND3, deactive** **ND3, turnover**

**Figure 10.** Model for subunit ND3 (yellow) and cryo-EM density (grey mesh) of subunit ND3 in the deactive state (left) and under turnover conditions (right). The central part of the long loop connecting TMH1 and 2 is disordered.

DOI: https://doi.org/10.7554/eLife.39213.018

prevented by a bound inhibitor molecule. However, this can now be excluded, as in our cryo-EM structure of *Y. lipolytica* complex I no inhibitor was present. Yet unfolding of the ubiquinone reduction site was not observed, at least not to an extent comparable to mammalian complex I . The species-dependent degrees of disorder in this critical region of complex I may reflect the significantly higher energy barrier for the A/D transition in bovine complex I as compared to *Y. lipolytica* (*Maklashina et al., 2003*). It is interesting to note that in mouse complex I the interface between accessory subunits NDUFA5 and NDUFA10 changes during the A/D transition; strong contacts between the two subunits seem to selectively stabilize the A form (*Agip et al., 2018*). This effect can be excluded for the *Y. lipolytica* complex, because it does not have an NDUFA10 ortholog.

**Table 3.** Lipid content of typical complex I preparation

| Lipid | nmol lipid/nmol complex I |
| --- | --- |
| phosphatidylcholine | 19.3 |
| lyso-phosphatidylcholine | 0.1 |
| phosphatidylethanolamine | 13.3 |
| phosphatidylserine | 0.5 |
| phosphatidylinositol | 11.8 |
| cardiolipin | 21.7 |
| $\Sigma$ | 66.7 |

DOI: https://doi.org/10.7554/eLife.39213.017

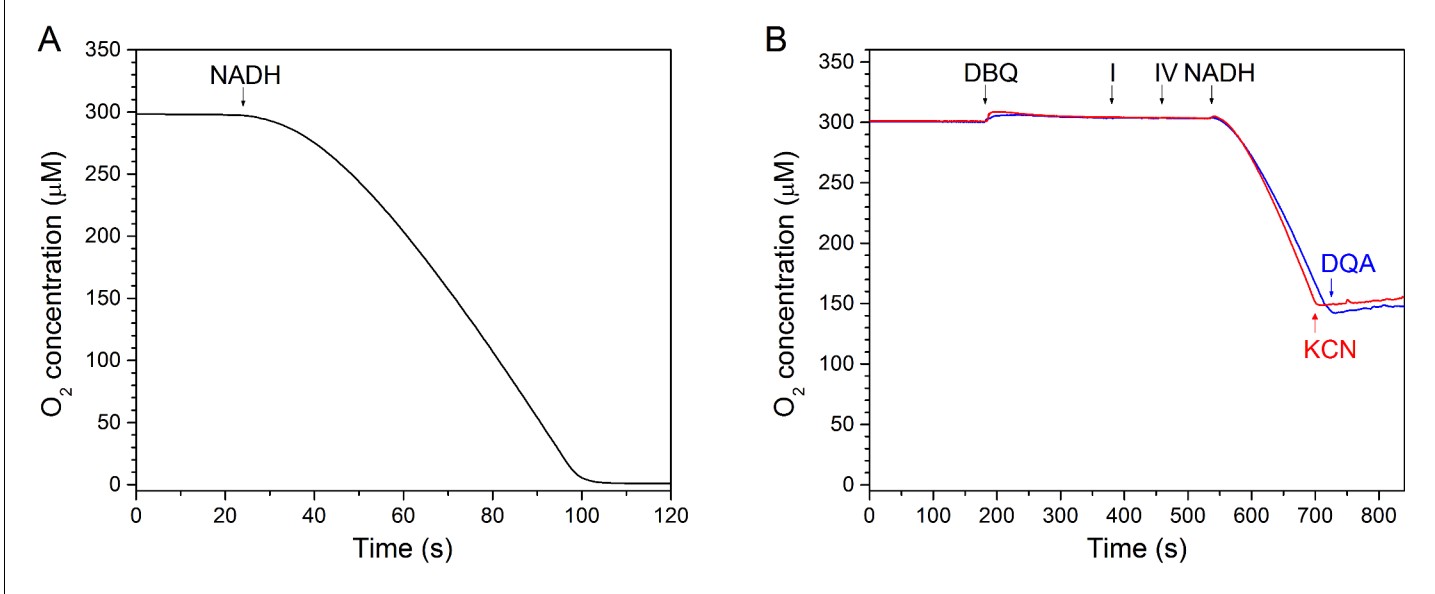

**Figure 11.** In vitro assay of a minimal respiratory chain of complex I and *bo₃*-type ubiquinol oxidase. (**A**) Assay conducted at the substrate concentration used for cryo-EM sample preparation (2 µM complex I , 1 µM oxidase, 2 mM NADH and 200 µM DBQ at 18°C. The reaction was started by addition of NADH. (**B**) Inhibition of complex I by DQA (blue) and of the *Vitreoscilla* oxidase by CN⁻ (red).

DOI: https://doi.org/10.7554/eLife.39213.019

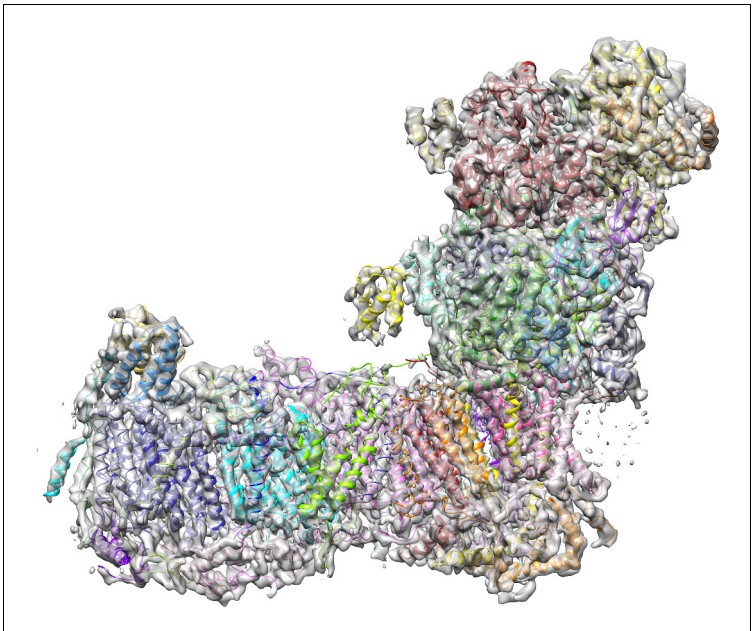

**Figure 12.** Complex I in the deactive state and under turnover conditions. The model for complex I in the deactive state (colour) is overlaid with the cryo-EM density (grey) for complex I under turnover conditions. There are no differences in overall structure, so there is no indication that the matrix arm moves relative to the membrane arm during turnover. Occupation and conformational changes of the substrate binding sites are shown in *Figure 13*.

DOI: https://doi.org/10.7554/eLife.39213.020

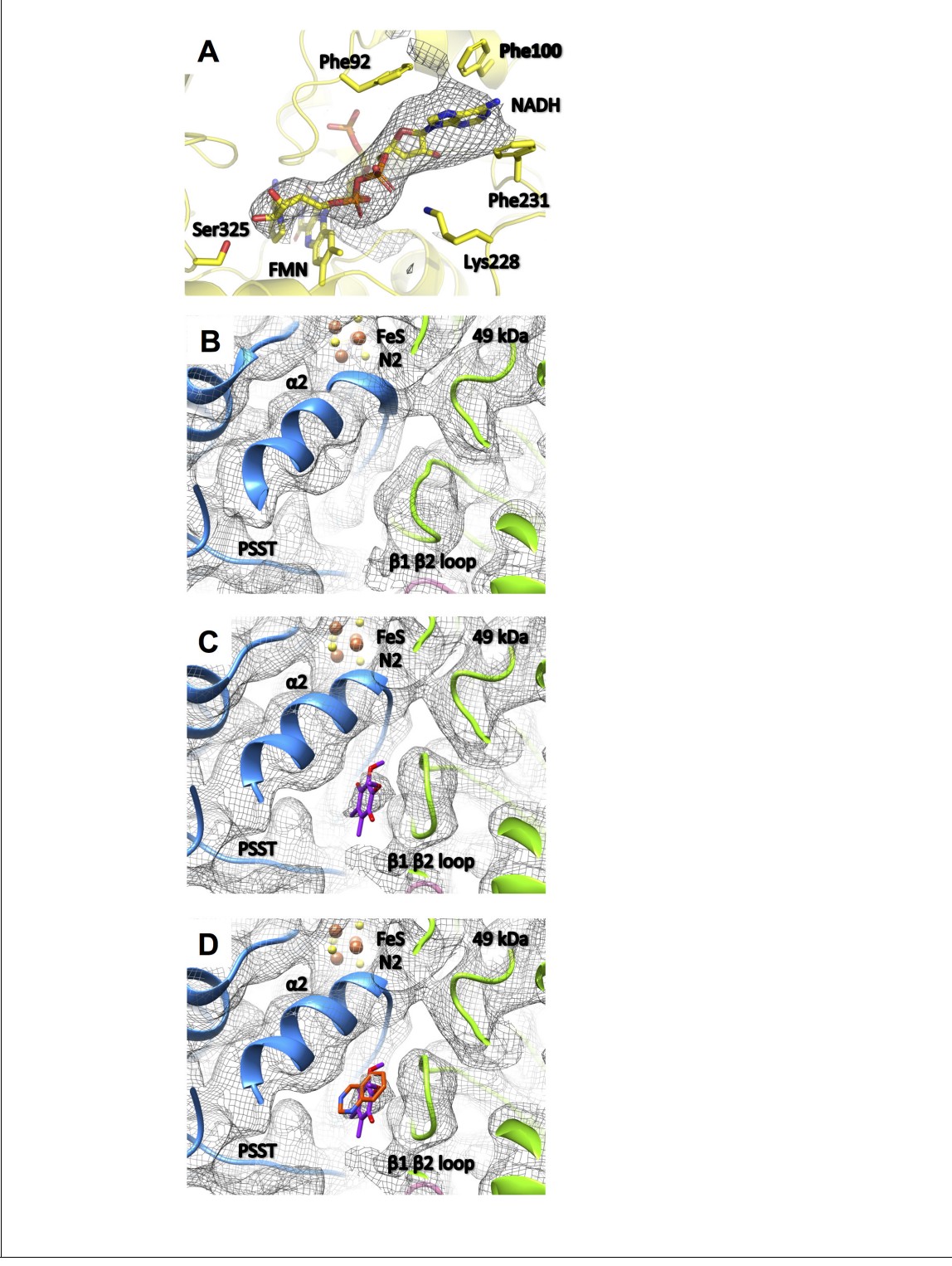

**Figure 13.** Substrate binding sites of complex I . (A) Under steady-state turnover conditions NADH (mesh, cryo-EM density) binds to the FMN cofactor and residues of the 51 kDa subunit; (B) ubiquinone binding site in the deactive state (mesh, cryo-EM density; 49 kDa subunit, green; PSST subunit, blue) and under steady-state turnover (C). The ubiquinone headgroup (purple) binds between the β1-β2 loop of the 49 kDa subunit and helix α2 of PSST. (D) This binding site overlaps with the position of the toxophore of decyl-quinazolineamine (orange) that was modelled based on anomalous diffraction of brominated inhibitor derivatives in the X-ray structure of *Y. lipolytica* complex I (*Zickermann et al., 2015*).
DOI: https://doi.org/10.7554/eLife.39213.021

eLIFE Research article                                    Structural Biology and Molecular Biophysics

**Figure 14.** Alternating binding positions of ubiquinone support a two-state stabilization change mechanism for respiratory complex I  (*Brandt, 2011*).
(**A**) ubiquinone and Tyr144 of the 49 kDa subunit (stick representation) and the β1-β2 loop of the 49 kDa and helix α2 and FeS cluster N2 of the PSST
*Figure 14 continued on next page*

*Figure 14 continued*
subunit in the ubiquinone binding pocket of *Y. lipolytica* complex I  (green) were superimposed on ubiquinone and the corresponding structures in *T. thermophilus* (grey). The position of ubiquinone in *T. thermophilus* (PDB ID: 4HEA) was fitted according to *Figure 4* in (*Baradaran et al., 2013*). The position of ubiquinone in *T. thermophilus* is assigned to the E-state (E) while the position of ubiquinone determined in our study is assigned to the P-state (P). (B) Electron transfer from iron-sulfur cluster N2 occurs in the E-state (grey), while ubiquinone intermediates are protonated in the P-state (green). The stabilization of negatively charged redox intermediates of ubiquinone drive the transition from the E- to the P-state, changing the binding site for the ubiquinone headgroup. This would create conformational and electrostatic strain in the loops lining the ubiquinone binding pocket. (C) The strain provides the energy for a power stroke transmitted through a chain of titratable residues (orange) into the membrane arm, where it drives the proton pump modules (red dots) (*Zickermann et al., 2015*).
DOI: https://doi.org/10.7554/eLife.39213.022

Ubiquinone reduction releases the energy for proton translocation. The ubiquinone reduction site therefore has to play a central role in energy conversion. Our data provide direct evidence for our earlier proposal of a second binding site for ubiquinone within the substrate binding pocket of complex I  (*Zickermann et al., 2015*), which becomes dominant during steady-state turnover. As compared to the D-form, under turnover conditions the only notable difference observed nearby was a reorganization of the β1-β2-loop of the 49 kDa subunit (*Figure 13B,C*). The critical loop connecting TMH1 and TMH2 of ND3 is flexible in both states (*Figure 10*). These findings support our proposed integrated functional model, which suggests that the structural changes associated with the A/D transition and the catalytic cycle of complex I  are in fact closely linked (*Zickermann et al., 2015*). According to this model, the D-form represents the enzyme locked in an intermediate state of the catalytic cycle termed P-state in the proposed two-state stabilization-change mechanism (*Brandt, 2011*) (*Figure 14B*). The E-state would be characterized by a ubiquinone binding site closer to iron-sulfur cluster N2 (*Figure 14A*), which so far was observed only in the oxidized bacterial enzyme (*Baradaran et al., 2013*) that does not undergo the A/D transition. A larger distance between the ubiquinone headgroup and its primary electron donor is in principle consistent with the impaired reduction in the P-state, as predicted by the mechanistic model. The observed 12 Å distance between the ubiquinone headgroup and cluster N2 in the P-state is within the <14 Å range that would still allow efficient electron transfer according to the 'Moser-Dutton ruler' (*Moser et al., 2010*) commonly used to estimate electron tunnelling rates between redox centres in proteins. However, in the present case, electron transfer is linked to protonation, and therefore other parameters, such as reorganization energy and packing density, have to be taken into consideration. These parameters do not depend simply on distance but on the local protein environment. They are likely to slow down electron transfer sufficiently for the mechanistic model we propose, even at the short distance observed for the P-state. Further studies and higher-resolution structures will be required to resolve this issue.

In the catalytic cycle, the E-state is predicted to be short-lived under the uncoupled steady-state turnover conditions applied here, explaining the observed predominance of the P-state. Once bound to the E-site, ubiquinone is rapidly reduced by iron sulfur-cluster N2, which essentially remains reduced on the time scale of catalytic turnover (*Krishnamoorthy and Hinkle, 1988*) (*Figure 14B*). The resulting highly unstable semiquinone then rapidly triggers the E → P conversion, creating the alternate binding site for the ubiquinone headgroup (*Zickermann et al., 2015*). These rearrangements are driven by thermodynamic stabilization of the charged ubiquinone intermediate that provides the energy for the electrostatic power stroke (*Brandt, 2011*). The power stroke is then transmitted through the chain of protonable residues into the membrane arm, where it ultimately drives proton pumping (*Figure 14C*). Subsequently, neutral semiquinone produced by protonation in the P-state allows the system to return to the E-state. The semiquinone then picks up the second electron to trigger another power stroke driven by the stabilization of the ubiquinol anion (*Figure 14B*).

In summary, our cryo-EM structure of mitochondrial complex I  under turnover conditions corroborates several predictions of the proposed two-state stabilization change mechanism and how it relates to the regulatory A/D transition. Our new structure provides strong support for our integrated functional model that describes the structural basis of energy conversion and regulation in respiratory complex I .

# Materials and methods

## Key resources table

| Reagent type (species) or resource | Designation | Source or reference | Identifiers | Additional information |
|---|---|---|---|---|
| Strain, strain background (*Y. lipolytica* GB20) | Δmus51, lys1⁻, leu2⁻, ura3⁻, 30Htg2, ndh2i | PMID 24706851 | | |
| Strain, strain background (*E. coli* CLY) | derived from C43(DE3), ΔcyoBCD::kan | PMID 17267395 | | Prof. Robert B. Gennis, University of Illinois |
| Genetic reagent (pET17b-V14) | cyoABCDE in pET 17b | this work | | Dr. Hao Xie, MPI for Biophysics, Frankfurt |
| Chemical compound, drug | n-Dodecyl β-maltoside | Glycon Biochemicals GmbH | Cat. # D97002-C | |
| Chemical compound, drug | Decylubiquinone | Sigma-Aldrich/Merck | Cat. # D7911 | |
| Chemical compound, drug | β-Nicotinamide adenine dinucleotide | Sigma-Aldrich/Merck | Cat. # N8129 | |
| Chemical compound, drug | Asolectin from soybean | Sigma-Aldrich/Merck | Cat. # 11145 | |
| Chemical compound, drug | CHAPS, Anagrade | Anatrace | Cat. # C316 | |
| Software, algorithm | Coot | PMID: 15572765 | RRID: SCR_014222 | |
| Software, algorithm | CTFFIND4 | PMID:26278980 | | |
| Software, algorithm | Gctf | PMID:26592709 | | |
| Software, algorithm | MolProbity | PMID: 20057044 | RRID: SCR_14226 | |
| Software, algorithm | MotionCor2 | PMID: 28250466 | | |
| Software, algorithm | Phenix | PMID: 20124702 | RRID: SCR_014224 | |
| Software, algorithm | PyMOL | Schrödinger, LLC | RRID: SCR_000305 | |
| Software, algorithm | RELION | PMID: 27845625 | RRID: SCR_016274 | |
| Software, algorithm | UCSF Chimera | PMID: 15264254 | RRID: SCR_004097 | |
| Software, algorithm | TMHMM Server | PMID: 11152613 | RRID: SCR_014935 | |

## Purification and characterization of respiratory complex I from *Yarrowia lipolytica*

Cells were grown in a 10-liter Biostat C fermenter (Braun, Germany) under high aeration and broken with a glass bead mill cell-disintegrator (Euler Biotechnologie, Germany). Mitochondrial membranes were isolated by differential centrifugation in 20 mM MOPS, pH 7.2, 600 mM sucrose, 1 mM EDTA. Complex I was purified in dodecyl maltoside (DDM) by His-tag affinity chromatography and gel filtration as described (*Kashani-Poor et al., 2001*) but with reduced detergent concentration in column buffers (0.025%) to preserve native lipids that are essential for enzyme activity. Under our standard assay conditions (60 μM DBQ, no added lipid or detergent) the preparation had an activity of 1.8 $\mu Mol^{-1}mg^{-1}min^{-1}$, increasing to 5.9 $\mu Mol^{-1}mg^{-1}min^{-1}$ upon lipid reactivation (*Dröse et al., 2002*). Using the assay conditions established for bovine complex I (200 μM DBQ, 0.15% asolectin, 0.15% CHAPS) (*Blaza et al., 2018*) the activity of the preparation was 13.9 $\mu Mol^{-1}mg^{-1}min^{-1}$.

## Determination of lipids by mass spectrometry

Lipids of human erythrocytes were extracted with an MTBE protocol according to Matyash *et al.* (*Matyash et al., 2008*). Lipid extracts were resuspended in 1000 µl CHCl$_3$/MeOH 1:1 and 12:0/13:0 PC, 17:0/20:4 PC, 14:1/17:0 PC, 21:0/22:6 PC, 12:0/13:0 PI, 17:0/20:4 PI, 14:1/17:0 PI, 21:0/22:6 PI (2 µM each) 17:1 LPC, (1.5 µM each), 12:0/13:0 PE, 17:0/20:4 PE, 14:1/17:0 PE, 21:0/22:6 PE, 12:0/13:0 PS, 17:0/20:4 PS, 14:1/17:0 PS, 21:0/22:6 PS (3 µM each), 14:1/14:1/14:1/15:1 CL (2.97 µM), 15:0/15:0/15:0/16:1 CL (2.70 µM), 22:1/22:1/22:1/14:1 CL (2.36 µM) and 24:1/24:1/24:1/14:1 CL (2.27 µM) were added as internal standard. 2 µl were injected on a Waters BEH C8, 100 × 1 mm, 1.7 µm HPLC column used with an Ultimate 3000 UHPLC (Thermo Scientific, USA). Solvent A was water with 1% ammonium acetate and 0.1% formic acid, and solvent B was acetonitrile/2-propanol 5:2 with 1% ammonium acetate and 0.1% formic acid. Gradient elution started at 50% mobile phase B, rising to 100% B over 40 min; 100% B were held for 10 min and the column was re-equilibrated with 50% B for 8 min before the next injection. The flow rate was 150 µl/min.

Data acquisition was performed according to Triebl *et al.* (*Triebl et al., 2017*) by Orbitrap-MS (LTQ-Orbitrap, Thermo Scientific) full scan in preview mode at a resolution of 100,000 and <2 ppm mass accuracy with external calibration. The spray voltage was set to 4500 V and the capillary temperature was at 300°C. From the FT-MS preview scan the 10 most abundant *m/z* values were picked in data dependent acquisition (DDA) mode, fragmented in the linear ion-trap analyser and ejected at nominal mass resolution. Normalized collision energy was set to 50%, the repeat count was two and the exclusion duration was 10 s. Data were analysed by Lipid Data Analyzer, a custom-developed software tool described in detail by Hartler *et al.* (*Hartler et al., 2011*; *Hartler et al., 2017*) (http://genome.tugraz.at/lda/lda_download.shtml).

## Construction of the expression vector of the cytochrome *bo*$_3$ ubiquinol oxidase

Genomic DNA of *Vitreoscilla* sp. C1 was isolated using the G-spin genomic extraction kit (iNtRON Biotechnology, South Korea). A 4,836 bp DNA fragment containing the *cyoABCDE* operon was amplified by PCR using primers Vbo3-NdeI und Vbo3-HindIII. The PCR product was digested with NdeI and HindIII endonucleases and ligated into the same site of pET-17b (Novagen, Germany), resulting in pET-17b-Vbo3. A TEV cleavage site and a deca-histidine tag was introduced at the C-terminus of the CyoA subunit by the InFusion ligation-independent cloning method (Clontech, USA), for purification by metal affinity chromatography. Simultaneously, an artificial ribosomal binding site was inserted upstream of the translational start site of the *cyoB* gene. For InFusion Cloning, the primer pairs Vbo3-10/Vbo3-15 and Vbo3-19/Vbo3-20 were used. The resulting final construct pET17b-V14 was verified by sequencing and introduced into the *E. coli* expression strain. The primer sequences are listed in *Table 4*.

## Expression of the cytochrome *bo*$_3$ ubiquinol oxidase

Expression vector pET17b-V14 was transformed into *E. coli* strain CLY (kindly provided by Prof. Robert B. Gennis, University of Illinois), which lacks the endogenous cytochrome *bo*$_3$ ubiquinol oxidase. A single colony was used to inoculate 50 ml of LB medium supplemented with 50 µg/mL kanamycin and 100 µg/mL carbenicillin. This pre-culture was grown at 37°C overnight and used to inoculate 2 liters of LB medium. The main culture was incubated at 30°C and 180 rpm until the optical density (OD) at 600 nm reached 0.6 – 0.8. Production of *Vitreoscilla bo*$_3$ ubiquinol oxidase was induced by addition of 0.5 mM isopropyl-β-D-thiogalactopyranoside (IPTG) and cultures were grown for 6 hr. Cells were harvested by centrifugation (10,500 × *g*, 4°C, 20 min) and resuspended in ice-cold resuspension buffer (50 mM potassium phosphate, pH 8.3, 5 mM MgCl$_2$, 1 mM Pefabloc, DNase I) at a ratio of 5 ml of buffer per 1 g of wet cells. Cells were disrupted four times by passing through an M-110LA microfluidizer (Microfluidics) on ice at 8,000 psi. Cell debris was removed by centrifugation (14,000 × *g*, 4°C, 1 hr). Membrane vesicles were collected from the supernatant by ultracentrifugation (214,000 × *g*, 4°C, 3 hr), flash-frozen in liquid nitrogen and stored at −80°C prior to solubilization.

**Table 4.** Oligonucleotides used in this work.

| Oligonucleotides | Sequence (5'−3') |
|---|---|
| Vbo3-NdeI[a] | GCGCATATGAAGCAGATGATTCAGGTC |
| Vbo3-HindIII[a] | GGGAAGCTTC AAAAATAAATATGCGGCAAC |
| Vbo3-10 | TAATCTATGTTAGGTAAACTCGATTGG |
| Vbo3-15 | ATTTCCTCCTGCAGCAGATGCAGCAAC |
| Vbo3-19[b] | GCTGCAGGAGGAAATGAAAACCTGTACTTTCAAGGTCATCACCATCACCATCAC CATCACCATCACTAAGCTGCATCTGCTGCAGGAGGAAATTAATCTATGTTAGGT |
| Vbo3-20[b] | ACCTAACATAGATTAATTTCCTCCTGCAGCAGATGCAGCTTAGTGATGGTGATG GTGATGGTGATGGTGATGACCTTGAAAGTACAGGTTTTCATTTCCTCCTGCAGC |

*Restriction enzymes sites are underlined.

[‡]The nucleotide sequences encoding the TEV cleavage site and the deca-histidine tag are shown in red and blue, respectively. The artificial intergenic region containing the *Vitreoscilla* ribosomal binding site is shown in magenta.

DOI: https://doi.org/10.7554/eLife.39213.023

## Purification of the cytochrome $bo_3$ ubiquinol oxidase

Crude membranes were resuspended in solubilization buffer (50 mM potassium phosphate, pH 8.3, 100 mM NaCl, 1 mM Pefabloc) at a ratio of 10 ml of buffer per 1 g of membrane. The total protein concentration was determined by the BCA assay (Pierce, USA) according to manufacturer's instructions. Membrane proteins were solubilized by moderate stirring of resuspended membranes with 2.5 mg DDM (GLYCON Biochemicals, Germany) per mg of protein at 4°C for 1 hr. The insoluble membrane fraction was removed by ultracentrifugation (214,000 × $g$, 4°C, 1 hr) and the supernatant containing solubilized membrane proteins was filtered through a 0.45 µm polyethersulfone (PES) membrane.

*Vitreoscilla bo$_3$* ubiquinol oxidase was purified in three chromatographic steps using an ÄKTA-purifier system (GE Healthcare, USA), including Ni$^{2+}$-NTA affinity, Q-Sepharose anion exchange and Superdex 200 gel filtration chromatography. Prior to affinity capture, imidazole was added to the solubilized membrane solution to a final concentration of 20 mM. Solubilized protein was loaded onto a Ni$^{2+}$-NTA column, equilibrated with 50 mM potassium phosphate, pH 8.3, 100 mM NaCl, 20 mM imidazole, 0.05% DDM. The column was washed with equilibration buffer until the $A_{280}$ and $A_{410}$ returned to baseline levels. Bound protein was eluted with Ni$^{2+}$-NTA elution buffer (50 mM potassium phosphate, pH 8.3, 100 mM NaCl, 150 mM imidazole, 0.05% DDM). Eluate fractions were pooled and diluted two-fold with 50 mM potassium phosphate, pH 8.3, 0.05% DDM, and loaded onto a Q-Sepharose high performance column pre-equilibrated with 20 mM Tris-HCl, pH 7.5, 0.03% DDM. After extensive washing, the protein was eluted with a linear 5 to 400 mM NaCl gradient in 20 mM Tris-HCl, pH 7.5, 0.03% DDM. Red-colored fractions were collected and concentrated using Amicon Ultra-15 concentrators (100K MWCO, Merck Millipore, Germany). The concentrated protein was purified further by chromatography on a Superdex 200 column equilibrated in 20 mM Tris-HCl, pH 7.5, 100 mM NaCl, 0.03% DDM. Peak fractions containing the cytochrome $bo_3$ ubiquinol oxidase from *Vitreoscilla* were collected and concentrated to a final concentration of ~200 µM and stored at −80°C.

## In vitro assay of a minimal respiratory chain

Oxygen consumption was determined polarographically with a Clark-type oxygen electrode (OX-MR; Unisense, Denmark) connected to a picoammeter (PA2000 Multimeter; Unisense, Denmark). Analog signals were converted into digital with an A/D converter (ADC-216; Unisense, Denmark) and then recorded with the software Sensor Trace Basic 2.1 supplied by the manufacturer.

In vitro assays of a minimal respiratory chain consisting of complex I and quinol oxidase were performed in 2 ml glass vials while stirring in a water bath at 18°C. The reaction vial was filled with 50 mM Tris-HCl, pH 7.5, 100 mM NaCl and 0.02% DDM, followed by the addition of 200 µM n-decylubiquinone (DBQ), 2 µM complex I and 1 µM cytochrome $bo_3$ ubiquinol oxidase (complex I V) to a final volume of 600 µl. The reaction was then initiated by adding 2 mM reduced nicotinamide

adenine dinucleotide (NADH), and inhibited by the addition of 2-decyl-4-quinazolinyl amine (DQA) or potassium cyanide (KCN).

## Cryo-EM

Deactive complex I was diluted to a final protein concentration of 2 µM in 20 mM Tris-HCl, pH 7.2, 100 mM NaCl, 1 mM EDTA and 0.025% DDM. For cryo-EM under turnover, 4 µM of complex I incubated with 400 µM DBQ was mixed 1:1 with 2 µM $bo_3$-type ubiquinol oxidase, tobacco mosaic virus as a spreading agent (0.13 mg/ml) and 4 mM NADH. 3 µl of the mixture was applied to freshly glow-discharged C-flat 1/1 holey carbon grids (Protochips, USA), automatically blotted (70% humidity, blot time 5 – 8 s, drain and wait time 0 s, blot force −2 a.u.) and flash-frozen in liquid ethane in an FEI Vitrobot ™ Mark IV (FEI NanoPort, the Netherlands). Cryo-EM data of inactive complex I were collected automatically with Leginon (*Suloway et al., 2005*) on a FEI Tecnai Polara at 300 kV equipped with a Gatan K2 direct electron detector operating in counting mode. Videos were collected at a total exposure of 60 e⁻/Å², at defocus values of −1.5 to −3.0 µm with a calibrated pixel size of 1.09 Å (200,000x). Cryo-EM images of complex I under turnover conditions were collected on a 300 kV FEI Titan Krios on a Falcon III direct electron detector operating in counting mode. Images were collected automatically with EPU software at a nominal magnification of 75,000x with a calibrated pixel size of 1.053 Å and a total exposure of 30 – 40 e⁻/Å², at a nominal defocus of −1.5 to −4.5 µm.

## Image processing

A set of 3110 micrographs of deactive complex I and 5650 micrographs of complex I under turnover conditions were motion-corrected and dose-weighted with MotionCor2 (*Zheng et al., 2017*) (*Figure 1A*). For deactive complex I, the micrograph-based contrast transfer function (CTF) was determined with CTFFIND4 (*Rohou and Grigorieff, 2015*). Particles were picked using Autopick within the RELION2.1 workflow (*Kimanius et al., 2016*), yielding 271,443 particles extracted in boxes of 456 × 456 pixels. Particles were subjected to initial reference-free two-dimensional (2D) classification in RELION2.1 to remove imperfect particles. Visual selection of class averages with interpretable features resulted in a dataset of 269,508 particles. These were used for 3D classification with a previous cryo-EM map of complex I from *Y. lipolytica* (*D'Imprima et al., 2016*) low-pass filtered to 40 Å as an initial model. A good class of 124,626 particles was used for auto-refinement and particle polishing in RELION2.1. After refinement the post-processing procedure implemented in RELION2.1 was applied to the final map for B-factor sharpening and resolution validation (*Chen et al., 2013*). The final map used for model building had a resolution of 4.3 Å, and was sharpened using an isotropic B-factor of −142 Å (*Figure 2A*, *Table 1*). Local map resolution was estimated with ResMap (http://resmap.sourceforge.net) (*Kucukelbir et al., 2014*) (*Figure 2*). To identify the ST1 subunit (*D'Imprima et al., 2016*), the map was subjected to a final round of 3D classification and a 3D class displaying an extra density protruding from the side of the matrix arm was used for a refinement which resulted in a map of 5.2 Å resolution from 36,723 particles (*Figure 1A*).

For complex I under turnover conditions, CTF parameters were estimated by Gctf (*Zhang, 2016*). Particles were picked automatically using 2D-class averages from the deactive complex I dataset for reference. The initial set of 273,581 particles extracted in boxes of 450 × 450 pixels was subjected to reference-free 2D classification in RELION2.1 to remove imperfect particles and $bo_3$-type ubiquinol oxidase. The remaining 257,951 particles were sorted by 3D classification and the map from inactive complex I low-pass filtered to 40 Å was used as initial model. The best class of 115,083 particles was subjected to auto-refinement and particle polishing in RELION2.1. The final map of 4.5 Å resolution was sharpened with an isotropic B-factor of −215 and the local resolution was estimated with ResMap.

## Model building

Homology models of individual subunits from *Y. lipolytica* complex I were created by SWISS-MODEL (*Biasini et al., 2014*) based on cryo-EM structures of *B. taurus* class 1 (PDB ID: 5LDW), *B. taurus* class 2 (PDB: 5LC5), *O. aries* (PDB ID: 5LNK), and the crystal structure of *Y. lipolytica* complex I (PDB ID: 4WZ7). Rigid body fitting into the cryo-EM map was performed with Chimera (*Pettersen et al., 2004*). The resulting model was adjusted to the density or manually built in COOT

(*Emsley and Cowtan, 2004*). Secondary structure predictions using the TMHMM server (*Krogh et al., 2001*) and well-resolved side chain densities guided model building (*Figure 3*). The model was refined in PHENIX (*Adams et al., 2010*) using phenix.real_space_refinement for six macro-cycles followed by several rounds of rebuilding in COOT. A quality check with MolProbity (*Chen et al., 2010*) indicated excellent stereochemistry with 85.83% of the non-glycine and non-proline residues found in the most-favoured region and 0.74% outliers (all-atom clashscore: 8.55). The model was cross-validated against overfitting by refinement in one half map (*Brown et al., 2015*) and showed no evidence of overfitting. Refinement and validation statistics are summarized in *Table 1*. Figures were drawn with Chimera (*Pettersen et al., 2004*) and PyMOL (The PyMOL Molecular Graphics System, Version 2.0, Schrödinger, LLC).

## Acknowledgments

The authors thank Arne Möller and Simone Prinz for help and advice during cryo-EM data collection, Paolo Lastrico for preparing figures, and Özkan Yildiz and Juan F Castillo-Hernández for computer support. We thank Martin Trötzmüller and Harald Köfeler (Core Facility Mass Spectrometry, Medical University of Graz) for the lipid analysis. Support by the Austrian Ministry for Education, Science and Research (JPI-HDHL Project No BMWFW-10.420/0005 W/V/3 c/2017 and HSRSM Grant Omics Center Graz, BioTechMed-Graz) is gratefully acknowledged.

## Additional information

### Competing interests

Werner Kühlbrandt: Reviewing editor, eLife. The other authors declare that no competing interests exist.

### Funding

| Funder | Grant reference number | Author |
| --- | --- | --- |
| Excellence Initiative of the German Federal and State Governments | EXC 115 | Ulrich Brandt Werner Kühlbrandt Volker Zickermann |
| Netherlands Organization for Scientific Research | TOP grant 714.017.004 | Ulrich Brandt |
| Deutsche Forschungsgemeinschaft | ZI 552/4-1 | Volker Zickermann |

The funders had no role in study design, data collection and interpretation, or the decision to submit the work for publication.

### Author contributions

Kristian Parey, Investigation, Methodology, Writing—review and editing; Ulrich Brandt, Conceptualization, Writing—original draft, Writing—review and editing; Hao Xie, Investigation, Cloned and purified the ubiquinol oxidase, Characterized and optimized the substrate regenerating system together with K.P; Deryck J Mills, Validation, Investigation; Karin Siegmund, Investigation, Prepared and characterized complex I ; Janet Vonck, Validation, Investigation, Writing—review and editing; Werner Kühlbrandt, Conceptualization, Resources, Supervision, Writing—review and editing; Volker Zickermann, Conceptualization, Supervision, Writing—original draft, Writing—review and editing

### Author ORCIDs

Kristian Parey https://orcid.org/0000-0002-4842-6479
Janet Vonck https://orcid.org/0000-0001-5659-8863
Werner Kühlbrandt https://orcid.org/0000-0002-2013-4810
Volker Zickermann https://orcid.org/0000-0001-8461-8817

Decision letter and Author response
Decision letter https://doi.org/10.7554/eLife.39213.044
Author response https://doi.org/10.7554/eLife.39213.045

# Additional files

## Supplementary files

• Supplementary file 1. Coordinate file of complex I under steady-state turnover conditions with NADH and ubiquinone binding sites occupied (compare EMD-4385). Please note that the position of the ubiquinone head group was identified but that the precise orientation of the molecule in the site remained ambiguous due to limited detail of the 4.5 Å resolution map.
DOI: https://doi.org/10.7554/eLife.39213.024

• Transparent reporting form
DOI: https://doi.org/10.7554/eLife.39213.025

## Data availability

The cryo-EM structure of deactive complex I has been deposited in the PDB with PDB ID 6GCS and the cryo-EM maps of deactive complex I and under turnover in the EMDB under accession numbers EMD-4384 and EMD-4385. The coordinate file of complex I under steady-state turnover conditions with NADH and ubiquinone binding sites occupied has been provided as Supplementary file 1. Please note that the position of the ubiquinone head group was identified but that the precise orientation of the molecule in the site remained ambiguous due to limited detail of the 4.5 Å resolution map.

The following datasets were generated:

| Author(s) | Year | Dataset title | Dataset URL | Database, license, and accessibility information |
|---|---|---|---|---|
| Kristian Parey, Janet Vonck | 2018 | cryo-EM structure of respiratory complex I from Yarrowia lipolytica | http://www.ebi.ac.uk/pdbe/entry/pdb/6gcs | Publicly available at the Electron Microscopy Data Bank (accession no: 6GCS) |
| Kristian Parey, Janet Vonck | 2018 | CRYOEM STRUCTURE OF RESPIRATORY COMPLEX I FROM YARROWIA LIPOLYTICA | http://www.ebi.ac.uk/pdbe/entry/emdb/EMD-4384 | Publicly available at the Electron Microscopy Data Bank (accession no: EMD-4384) |
| Kristian Parey, Janet Vonck | 2018 | CRYOEM STRUCTURE OF RESPIRATORY COMPLEX I FROM YARROWIA LIPOLYTICA UNDER TURNOVER | http://www.ebi.ac.uk/pdbe/entry/emdb/EMD-4385 | Publicly available at the Electron Microscopy Data Bank (accession no: EMD-4385) |

The following previously published datasets were used:

| Author(s) | Year | Dataset title | Dataset URL | Database, license, and accessibility information |
|---|---|---|---|---|
| Vinothkumar KR, Zhu J, Hirst J | 2016 | Structure of mammalian respiratory Complex I, class1 | https://www.rcsb.org/structure/5LDW | Publicly available at the RCSB Protein Data Bank (accession no: 5LDW) |
| Vinothkumar KR, Zhu J, Hirst J | 2016 | Structure of mammalian respiratory Complex I, class2 | https://www.rcsb.org/structure/5LC5 | Publicly available at the RCSB Protein Data Bank (accession no: 5LC5) |
| Fiedorczuk K, Letts JA, Kaszuba K, Sazanov LA | 2016 | Entire ovine respiratory complex I | https://www.rcsb.org/structure/5LNK | Publicly available at the RCSB Protein Data Bank (accession no: 5LNK) |

| Wirth C, Zicker-mann V, Brandt U, Hunte C | 2015 | Crystal structure of mitochondrial NADH:ubiquinone oxidoreductase from Yarrowia lipolytica | https://www.rcsb.org/structure/4WZ7 | Publicly available at the RCSB Protein Data Bank (accession no: 4WZ7) |
| Baradaran R, Berrisford JM, Minhas GS, Sazanov LA | 2013 | Crystal structure of the entire respiratory complex I from Thermus thermophilus | https://www.rcsb.org/structure/4HEA | Publicly available at the RCSB Protein Data Bank (accession no: 4HEA) |

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
