## [Decision Letter]

Thank you for submitting your article "The cryo-EM structure of respiratory complex I at work" for consideration by *eLife*. Your article has been reviewed by three peer reviewers, including Sjors HW Scheres as the Reviewing Editor and Reviewer #1, and the evaluation has been overseen by Richard Aldrich as the Senior Editor. The following individuals involved in review of your submission have agreed to reveal their identity: Marten Wikstrom (Reviewer #2); Vinoth Kumar (Reviewer #3).

The reviewers have discussed the reviews with one another and the Reviewing Editor has drafted this decision to help you prepare a revised submission.

Summary:

This manuscript describes the structures of Yeast Complex I in its deactive state and in a state where the enzyme is performing catalysis thereby supporting a two-state stabilization change mechanism. The structure (higher-resolution than the previous) of yeast complex I now joins other complex I structures from different species determined by cryo-EM in recent times. The manuscript merits publication as the yeast complex I has a few salient features (additional/different subunits – useful for understanding the evolution of the enzyme and the role of these subunits), and it represents the first observation of a ubiquinone derivative bound to the enzyme, identifying a site different from that inferred from other studies. The following suggestions would improve the manuscript.

Essential revisions:

1) The authors' presentation of a possible mechanism of proton pumping would do well by acknowledging that Euro et al., (2008) were the first to suggest that the negative charge of an anionic form of either semi-ubiquinone or ubiquinol would start an essentially electrostatic coupling mechanism that would spread from the site of Q reduction to the proton pump centers in the membrane domain.

2) The authors should give the edge-to-edge distance between the quinone in the observed binding site and the Fe/S center N2, and/or the tyrosine close to the latter. If this distance is significantly less than ~14Å some new thinking may be required.

3) The recent structures of mouse enzyme (Agip et al. 2018, NSMB), where the authors have imaged the complex in defined biochemical states shows that there might be distinct states in Complex I. On this context, I feel that in the current manuscript this should be discussed and not be biased in their interpretation that no large structural change occurs (the relative position of matrix/membrane arm). While it might still be true that large structural changes might not be required, the presence of detergent environment (though lipids are present) and the possibility that the ubiquinone might exit through a different pathway than the entry should be considered.

4) The authors say that they observed only one major class (Results section) indicating a homogenous state but the resolution they have obtained is lower than that reported for mammalian complex I. This kind of contradicts the statement in Discussion section as they argue that the unfolding (more appropriate will be disorder) and change in orientation in bovine enzyme is due to lesser stability when released from supercomplex. If the yeast enzyme was homogenous then the resolution should have been better! In addition, the mouse Complex I reveals a resolution of 3.3 Å with ~1/10th of particles used in the yeast enzyme preparation. The number of particles used for reconstruction for the current structures seem to imply that there is something happening during the biochemical procedure and or grid. This may warrant some discussion.

5) The authors make a point of their model being more complete than the previously available X-ray structure. But how can the authors be sure about assigning all the accessory sub-units at these resolutions? How much ambiguity still exists? Supplementary figures with zoomed views of density features that make each subunit's assignment unique should be added to remove doubts about ambiguity of the assignments. Or if some ambiguities remain, these should be mentioned clearly in the text.

6) Refinement of the atomic models is cumbersome at these resolutions. Overfitting is a real possibility. To show that the model is not overfitting the map, the authors should scramble and then re-refine the model in one of the half-maps, and subsequently calculate FSC curves of that refined model with both of the half-maps. A supplementary figure of these curves should be added to the paper. The curves should not be far from each other (which happens if the model overfits the data). If they are, the weight on the cryo-EM density relative to the restraints should be lowered, and the final model (which can still be refined in the combined, final map) should be re-refined with those weights.

7) Provided model refinement can be done without too much overfitting, the model for the turnover map could be refined and both the map and the model could be deposited. The refinement statistics should then also be added to the relevant table. This is because, the deactive state will not have the DBQ modelled and if a reader wants to study/probe/understand the binding site the only option is Figure 13 (as the authors have themselves found in Baradaran et al., 2013, where binding site is depicted in the figure but there is no map deposited).

8) A normalised difference map (at a given σ threshold) might be useful to show the density for DBQ. As there are no major changes this should come out very nicely.

9) It may be useful to show/define which of the states of mammalian complex I best matches with the yeast Complex I. There is an overlay of yeast X-ray and bovine deactive EM model in Figure 5C but its relevance is not mentioned either in the main text or the legend. I don't see a reason why this should be shown. Instead, an overlay of EM derived models of yeast deactive and mammalian (bovine or mouse) active/deactive might be informative.

---

## [Author Response]

Summary:This manuscript describes the structures of Yeast Complex I in its deactive state and in a state where the enzyme is performing catalysis thereby supporting a two-state stabilization change mechanism. The structure (higher-resolution than the previous) of yeast complex I now joins other complex I structures from different species determined by cryo-EM in recent times. The manuscript merits publication as the yeast complex I has a few salient features (additional/different subunits – useful for understanding the evolution of the enzyme and the role of these subunits), and it represents the first observation of a ubiquinone derivative bound to the enzyme, identifying a site different from that inferred from other studies. The following suggestions would improve the manuscript.Essential revisions:1) The authors' presentation of a possible mechanism of proton pumping would do well by acknowledging that Euro et al., (2008) were the first to suggest that the negative charge of an anionic form of either semi-ubiquinone or ubiquinol would start an essentially electrostatic coupling mechanism that would spread from the site of Q reduction to the proton pump centers in the membrane domain.

A central role of semiquinone for energy conversion was suggested before, but indeed Euro et al., were the first to propose electrostatic energy transfer to the membrane. We have added the reference to the manuscript.

2) The authors should give the edge-to-edge distance between the quinone in the observed binding site and the FeS center N2, and/or the tyrosine close to the latter. If this distance is significantly less than ~14Å some new thinking may be required.

The edge-to-edge distance between the fitted ubiquinone molecule and FeS cluster N2 is approximately 12 Å, while the corresponding distance for the alternative position found in Thermus thermophilus is approximately 10 Å. We are aware that according to the “Dutton ruler” the binding position for ubiquinone determined here under steady state turnover conditions is within the 14 Å limit generally accepted for efficient electron transfer. However, the “Dutton ruler” does not strictly apply in our situation, because it refers to electron tunneling rates between metal centers in proteins, whereas the reduction of an organic molecule (ubiquinone) involves redox chemistry, which would not have the same, simple distance dependence. We therefore do not see any contradiction between the 12 Å distance and our proposed model that assigns this binding position to the P state. In the Results section of the revised manuscript, we state the 12 Å distance between cluster N2 and ubiquinone:

“We conclude that the new density represents the headgroup of a bound ubiquinone substrate at a minimal edge-to-edge distance of approximately 12 Å from cluster N2. This site is different from the ubiquinone binding site reported for the bacterial enzyme from Thermus thermophilus, which engages a strictly conserved tyrosine adjacent to the terminal iron-sulfur cluster N2 and positions the quinone headgroup about 2 Å closer to cluster N2 (Baradaran et al., 2013) (Figure 14A).”

We also added a short paragraph to the Discussion section to explain our reasoning as follows:

“A larger distance between the ubiquinone headgroup and its primary electron donor is in principle consistent with the impaired reduction in the P-state, as predicted by the mechanistic model. […] They are likely to slow down electron transfer sufficiently for the mechanistic model we propose, even at the short distance observed for the P-state. Further studies and higher-resolution structures will be required to resolve this issue.”

3) The recent structures of mouse enzyme (Agip et al. 2018, NSMB), where the authors have imaged the complex in defined biochemical states shows that there might be distinct states in Complex I. On this context, I feel that in the current manuscript this should be discussed and not be biased in their interpretation that no large structural change occurs (the relative position of matrix/membrane arm). While it might still be true that large structural changes might not be required, the presence of detergent environment (though lipids are present) and the possibility that the ubiquinone might exit through a different pathway than the entry should be considered.

We agree that the very recent NSMB paper by Agip et al., deserves to be cited. Please note that this paper appeared after our manuscript was submitted. The major differences between the deactive forms of *Y. lipolytica* and bovine complex I were already discussed in our original manuscript. We can now generalize our conclusions on the mammalian enzyme, as the unfolding patterns of loops in the deactive forms of bovine and mouse complex I are very similar.

We did not intend to say that distinct states do not exist in complex I. In fact, we compare two different functional states. Concerning the question of large conformational changes, we find that the relative orientation of the two arms does not appear to change in *Y. lipolytica*, whereas such a movement is now consistently observed in two mammalian species. Therefore, we omitted our speculation that a change in relative orientation of the two arms might be a stability issue and related to the release of complex I from the supercomplex. Overall, the differences observed between bovine, mouse, and yeast complex I substantiate our previous conclusion that the species-dependent degrees of disorder reflect the significantly higher energy barrier for the A/D transition in mammalian complex I. Interestingly, a comparison of our structure to that from mouse in the active and deactive form offers a straightforward explanation for this remarkable difference. We added a short paragraph to the Discussion to point this out:

“It is interesting to note that in complex I from mouse the interface between accessory subunits NDUFA5 and NDUFA10 changes during the A/D transition and strong contacts between the two subunits seem to selectively stabilize the A form (Agip et al., 2018). This effect can be excluded for the *Y. lipolytica* complex, because it does not have a NDUFA10 ortholog.”

We feel that in the absence of unequivocal data a further discussion of possible ubiquinone entry and exit pathways would be speculative and beyond the scope of this manuscript.

4) The authors say that they observed only one major class (Results section) indicating a homogenous state but the resolution they have obtained is lower than that reported for mammalian complex I. This kind of contradicts the statement in Discussion section as they argue that the unfolding (more appropriate will be disorder) and change in orientation in bovine enzyme is due to lesser stability when released from supercomplex. If the yeast enzyme was homogenous then the resolution should have been better! In addition, the mouse Complex I reveals a resolution of 3.3 Å with ~1/10th of particles used in the yeast enzyme preparation. The number of particles used for reconstruction for the current structures seem to imply that there is something happening during the biochemical procedure and or grid. This may warrant some discussion.

In contrast to mammalian complex I, we see only one major class in the data without variability of the angle between the two arms. There are several reasons why the resolution of our maps is less high than that of the mouse map. The data for the deactive map was recorded on the Polara electron microscope, which is less stable than the Krios and therefore the data are slightly less good. The turnover data were collected on a Krios, but image contrast was reduced by the high concentrations of substrates in the sample.

5) The authors make a point of their model being more complete than the previously available X-ray structure. But how can the authors be sure about assigning all the accessory sub-units at these resolutions? How much ambiguity still exists? Supplementary figures with zoomed views of density features that make each subunit's assignment unique should be added to remove doubts about ambiguity of the assignments. Or if some ambiguities remain, these should be mentioned clearly in the text.

We added two supplementary figures to show that the assignments are unambiguous, and changed the paragraph on accessory subunits (see below).

6) Refinement of the atomic models is cumbersome at these resolutions. Overfitting is a real possibility. To show that the model is not overfitting the map, the authors should scramble and then re-refine the model in one of the half-maps, and subsequently calculate FSC curves of that refined model with both of the half-maps. A supplementary figure of these curves should be added to the paper. The curves should not be far from each other (which happens if the model overfits the data). If they are, the weight on the cryo-EM density relative to the restraints should be lowered, and the final model (which can still be refined in the combined, final map) should be re-refined with those weights.

We are pleased to add the result of this validation test in a new supplement to Figure 3. The curves do not indicate any overfitting.

7) Provided model refinement can be done without too much overfitting, the model for the turnover map could be refined and both the map and the model could be deposited. The refinement statistics should then also be added to the relevant table. This is because, the deactive state will not have the DBQ modelled and if a reader wants to study/probe/understand the binding site the only option is Figure 13 (as the authors have themselves found in Baradaran et al., 2013, where binding site is depicted in the figure but there is no map deposited).

We deposited the map and the model for the deactive form and the map for complex I under turnover conditions. Obviously, at 4.5 Å resolution the ubiquinone molecule cannot be fitted precisely. Since inexperienced readers will not be aware of this, we prefer not to deposit the model with fitted UQ in the PDB for the time being. The 4.5 Å resolution map has been supplied as Supplementary file 1.

8) A normalised difference map (at a given σ threshold) might be useful to show the density for DBQ. As there are no major changes this should come out very nicely.

The density for the Q in the difference map is weak because the site is either not fully occupied or the ubiquinone molecule is flexible. However, the Q site shows positive density at every stage of refinement for the structure under the turnover conditions, whereas this is not the case for the deactive state, indicating that the density is real. An overlay of the two maps is shown in Author response image 1.

**Author response image 1. respfig1:** Density for ubiquinone headgroup. View from the membrane arm into the ubiquinone binding pocket. Overlay of density maps (deactive, blue; turnover conditions, red; model for complex I under turnover conditions). Note that the blue map has slightly higher resolution and that the b1 b2 loop changes between the two states.

9) It may be useful to show/define which of the states of mammalian complex I best matches with the yeast Complex I. There is an overlay of yeast X-ray and bovine deactive EM model in Figure 5C but its relevance is not mentioned either in the main text or the legend. I don't see a reason why this should be shown. Instead, an overlay of EM derived models of yeast deactive and mammalian (bovine or mouse) active/deactive might be informative.

Figure 5C is an overlay of the yeast X-ray structure with our deactive yeast EM map, not with the bovine map. The point of the figure is to show that the EM and X-ray structures of the same complex are consistent. We have clarified this in the legend. Fitting yeast to mammalian complex I is complicated because of a rotation of the two arms relative to one another, so that neither of them match.